# MEMORY, BENCHMARK & ROBOTS: A BENCHMARK FOR SOLVING COMPLEX TASKS WITH REINFORCEMENT LEARNING

**Egor Cherepanov**[1,2], **Nikita Kachaev**[1,3], **Alexey K. Kovalev**[1,2], **Aleksandr I. Panov**[1,2]
[1]AXXX, [2]MIRIAI, [3]ITMO University AI Talent Hub
`cherepanov@axxx.tech`

## ABSTRACT

Memory is crucial for enabling agents to tackle complex tasks with temporal and spatial dependencies. While many reinforcement learning (RL) algorithms incorporate memory, the field lacks a universal benchmark to assess an agent's memory capabilities across diverse scenarios. This gap is particularly evident in tabletop robotic manipulation, where memory is essential for solving tasks with partial observability and ensuring robust performance, yet no standardized benchmarks exist. To address this, we introduce **MIKASA** (**M**emory-**I**ntensive **Sk**ills **A**ssessment **S**uite for **A**gents), a comprehensive benchmark for memory RL, with three key contributions: (1) we propose a comprehensive classification framework for memory-intensive RL tasks, (2) we collect **MIKASA-Base** – a unified benchmark that enables systematic evaluation of memory-enhanced agents across diverse scenarios, and (3) we develop **MIKASA-Robo**[1] – a novel benchmark of 32 carefully designed memory-intensive tasks that assess memory capabilities in tabletop robotic manipulation. Our work introduces a unified framework to advance memory RL research, enabling more robust systems for real-world use. **MIKASA is available at** **https://tinyurl.com/membenchrobots**.

## 1 INTRODUCTION

Many real-world problems involve partial observability ([Kaelbling et al., 1998](#)), where an agent lacks full access to the environment's state. These tasks often include sequential decision-making ([Chen et al., 2021](#)), delayed or sparse rewards, and long-term information retention ([Lampinen et al., 2021](#); [Parisotto et al., 2020](#)). One approach to tackling these challenges is to equip the agent with memory, allowing it to utilize historical information ([Meng et al., 2021](#); [Ni et al., 2021](#)). While there are well-established benchmarks in Natural Language Processing ([An et al., 2023](#); [Bai et al., 2023](#)), the evaluation of memory in reinforcement learning (RL) remains fragmented. Existing benchmarks, such as POPGym ([Morad et al., 2023](#)), DMLab-30 ([Hung et al., 2018](#)) and MemoryGym ([Pleines et al., 2023](#)), focus on specific aspects of memory utilization, as they are designed around particular problem domains.

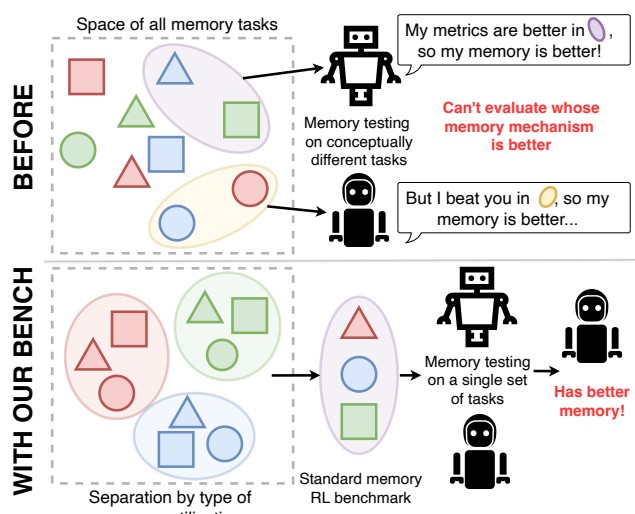

Figure 1: Systematic classification of problems with memory in RL reveals distinct memory utilization patterns and enables objective evaluation of memory mechanisms across different agents.

---

[1]`pip install mikasa-robo-suite`

Table 1: **MIKASA-Robo**: A benchmark comprising **32 memory-intensive robotic manipulation tasks** across 12 categories. Each task varies in difficulty and configuration modes. The table specifies episode timeout (T), the necessary information that the agent must memorize in order to succeed (Oracle Info), and task instructions (Prompt) for each environment. See Appendix K for details.

| Memory Task | Mode | Brief description of the task | T | Oracle Info | Prompt | Memory |
|---|---|---|---|---|---|---|
| ShellGame | Touch Push Pick | Memorize the position of the ball after some time being covered by the cups and then interact with the cup the ball is under | 90 | cup_with_ball_number | — | Object |
| Intercept | Slow Medium Fast | Memorize the positions of the rolling ball, estimate its velocity through those positions, and then aim the ball at the target | 90 | initial_velocity | — | Spatial |
| InterceptGrab | Slow Medium Fast | Memorize the positions of the rolling ball, estimate its velocity through those positions, and then catch the ball with the gripper and lift it up | 90 | initial_velocity | — | Spatial |
| RotateLenient | Pos PosNeg | Memorize the initial position of the peg and rotate it by a given angle | 90 | y_angle_diff | target_angle | Spatial |
| RotateStrict | Pos PosNeg | Memorize the initial position of the peg and rotate it to a given angle without shifting its center | 90 | y_angle_diff | target_angle | Spatial |
| TakeItBack-v0 | — | Memorize the initial position of the cube, move it to the target region, and then return it to its initial position | 180 | xyz_initial | — | Spatial |
| RememberColor | 3 \ 5 \ 9 | Memorize the color of the cube and choose among other colors | 60 | true_color_indices | — | Object |
| RememberShape | 3 \ 5 \ 9 | Memorize the shape of the cube and choose among other shapes | 60 | true_shape_indices | — | Object |
| RememberShape-AndColor | 3×2\3×3\ 5×3 | Memorize the shape and color of the cube and choose among other shapes and colors | 60 | true_shapes_info true_colors_info | — | Object |
| BunchOfColors | 3 \ 5 \ 7 | Remember the colors of the set of cubes shown simultaneously in the bunch and touch them in any order | 120 | true_color_indices | — | Capacity |
| SeqOfColors | 3 \ 5 \ 7 | Remember the colors of the set of cubes shown sequentially and then select them in any order | 120 | true_color_indices | — | Capacity |
| ChainOfColors | 3 \ 5 \ 7 | Remember the colors of the set of cubes shown sequentially and then select them in the same order | 120 | true_color_indices | — | Sequential |
| **Total: 32 tabletop robotic manipulation memory-intensive tasks in 12 groups** | | | | | | |

In contrast to classical RL, where benchmarks like Atari (Bellemare et al., 2013) and MuJoCo (Todorov et al., 2012) serve as universal standards, memory-enhanced agents are typically evaluated on custom environments developed alongside their proposals Table 2. This fragmented evaluation landscape obscures important performance variations across different memory tasks. For instance, an agent might excel at maintaining object attributes over extended periods while struggling with sequential recall challenges. Such task-specific strengths and limitations often remain hidden due to narrow evaluation scopes, underscoring the need for a comprehensive benchmark that spans diverse memory-intensive scenarios.

The challenge of memory evaluation becomes particularly evident in robotics. While some robotic tasks naturally involve partial observability, e.g. navigation tasks (Ai et al., 2022; Yadav et al., 2023), many studies artificially create partially observable scenarios from Markov Decision Processes (MDPs) (Kaelbling et al., 1996) by introducing observation noise or masking parts of the state space (Kurniawati, 2022; Lauri et al., 2023; Meng et al., 2021; Spaan, 2012). However, these approaches do not fully capture the complexity of real-world robotic challenges (Lauri et al., 2023), where tasks may require the agent to recall past object configurations, manipulate occluded objects, or perform multi-step procedures that depend heavily on memory. Such tasks include, for example, situations where a service robot needs to memorize occluded objects (e.g., a plate hidden under a towel) or where a home robot needs to accurately wipe the door of a microwave oven several times. Without memory, the robot wouldn't detect the plate in the first case, and in the second, it would wipe the door endlessly, unsure whether it has cleaned the area or if it's time to stop.

In this paper, we aim to address these challenges with the following four contributions:

1. **Memory Tasks Classification.** We propose a simple yet comprehensive framework that organizes memory-intensive tasks into four key categories. This structure enables systematic evaluation without added complexity (Figure 1), offering a clear guide for selecting environments that reflect core memory challenges in RL and robotics (Section 4).

2. **Memory-RL Benchmark.** We introduce **MIKASA-Base**, a Gymnasium-based (Towers et al., 2024) framework for evaluating memory-enhanced RL agents (Section 5).

3. **Robotic Manipulation Tasks.** We introduce **MIKASA-Robo**, an open-source benchmark (MIT license) comprising 32 robotic tasks that target specific memory-dependent skills in realistic settings (Section 6). We evaluate it using popular Online RL baselines (Section 6.2) as well as Visual-Language-Action (VLA) models (Section 6.4). Guidelines for customizing environments and configuring time horizons are provided in Appendix M.

4. **Robotic Manipulation Datasets.** We release datasets with expert quality trajectories for all 32 MIKASA-Robo memory-intensive tasks to support Offline RL research (see Appendix C), and conduct extensive evaluations using a range of Offline RL baselines (Section 6.3).

## 2 RELATED WORKS

Multiple RL benchmarks are designed to assess agents' memory capabilities. DMLab-30 (Hung et al., 2018) provides 3D navigation and puzzle tasks, focusing on long-horizon exploration and spatial recall. PsychLab (Leibo et al., 2018) extends DMLab by incorporating tasks that probe cognitive processes, including working memory. MiniGrid and MiniWorld (Chevalier-Boisvert et al., 2023) emphasize partial observability in lightweight 2D and 3D environments, while Mini-Hack (Samvelyan et al., 2021) builds on NetHack (Küttler et al., 2020), offering small roguelike scenarios that require both short- and long-term memory. BabyAI (Chevalier-Boisvert et al., 2019) combines natural language instructions with grid-based tasks, requiring memory for multi-step command execution. POP-Gym (Morad et al., 2023) standardizes memory evaluation with tasks ranging from pattern-matching puzzles to complex sequential decision-making.

Table 2: Key memory-intensive environments from the reviewed studies for evaluating agent memory. The Atari (Bellemare et al., 2013) environment with frame stacking is included to illustrate that many memory-enhanced agents are tested solely in MDP. Benchmark first introduced in the same work . Benchmark is open-sourced.

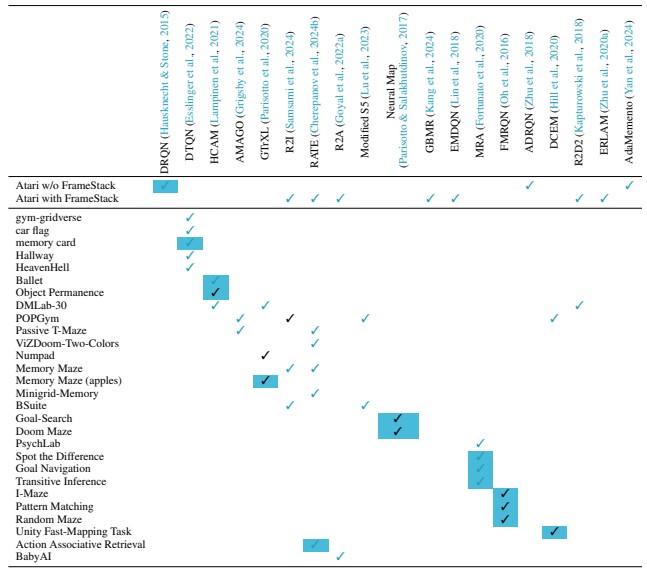

BSuite (Osband et al., 2020) offers a suite of carefully designed experiments that test core RL capabilities, including memory, through controlled tasks on exploration, credit assignment, and scalability. Memory Gym (Pleines et al., 2023) offers a suite of 2D grid environments with partial observability, designed to benchmark memory capabilities in decision-making agents, including endless versions of tasks for evaluating memory over extremely long time intervals. Memory Maze (Pasukonis et al., 2022) presents 3D maze navigation tasks that require memory to solve efficiently.

While these benchmarks offer valuable insights into memory mechanisms, they generally focus on abstract puzzles or navigation tasks. However, none of them fully encompass the broad range of memory utilization scenarios an agent may encounter, and the tasks themselves often differ fundamentally across benchmarks, making direct comparison of memory-enhanced agents difficult. In the robotics domain, memory requirements become particularly challenging due to the physical nature of manipulation tasks. Unlike abstract environments, robotic manipulation involves complex physical interactions and multi-step procedures demanding both spatial and temporal memory. Existing memory-intensive benchmarks, while useful for diagnostic purposes, struggle to capture these domain-specific challenges. The physical control and object interaction inherent in manipulation tasks introduce additional complexities not addressed by traditional memory evaluation frameworks.

Efforts have been made to classify memory-intensive environments by specific attributes. For example, Ni et al. (2023) divides them into memory/credit assignment based on temporal horizons. Yue et al. (2024) proposes memory dependency pairs to model how past events influence current decisions, aiding imitation learning in partially observable tasks. Cherepanov et al. (2024a) defines agent memory types: long-term vs. short-term (based on context length), and declarative vs. procedural (based on environments and episodes), and formalizes memory-intensive environments. Leibo et al. (2018) instead adapts tasks from cognitive psychology and psychophysics to evaluate agents on human cognitive benchmarks. While these classifications highlight aspects of memory, they overlook physical dimensions in robotics. The link between physical interaction and memory remains underexplored, motivating a framework for spatio-temporal memory in real-world tasks.

Concurrent with our work Fang et al. (2025) also proposed MemoryBench, a benchmark for memory-intensive manipulation consisting of only three tasks designed to access only one type of memory, spatial memory. This benchmark is based on RLBench (James et al., 2020), which does not allow efficient parallelization of training.

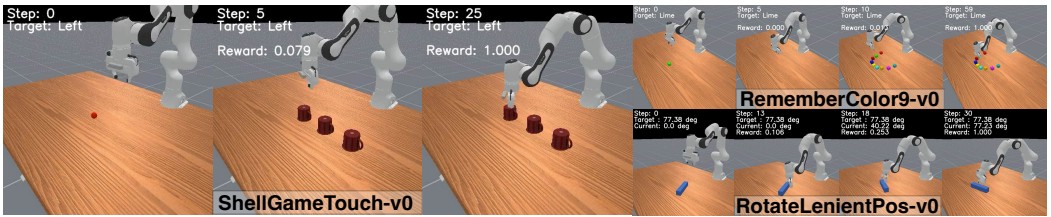

Figure 2: Illustration of demonstrative memory-intensive tasks execution from the proposed MIKASA-Robo benchmark. The `ShellGameTouch-v0` task requires the agent to memorize the ball's location under mugs and touch the correct one. In `RememberColor9-v0`, the agent must memorize a cube's color and later select the matching one. In `RotateLenientPos-v0`, the agent must rotate a peg while keeping track of its previous rotations.

## 3 BACKGROUND

### 3.1 PARTIALLY OBSERVABLE MARKOV DECISION PROCESS

Partially Observable Markov Decision Process (POMDP) (Kaelbling et al., 1996) extend MDP to account for partial observability, where an agent observes only noisy or incomplete information about the true environments state. POMDP defined by a tuple $(S, A, T, R, \Omega, O, \gamma)$, where: $S$ is the set of states representing the complete environment configuration; $A$ is the action space; $T(s'|s, a) : S \times A \times S \to [0, 1]$ is the transition function defining the probability of reaching state $s'$ from state $s$ after taking action $a$; $R(s, a) : S \times A \to \mathbb{R}$ is the reward function specifying the immediate reward for taking action $a$ in state $s$; $\Omega$ is the observation space containing all possible observations; $O(o|s, a) : S \times A \times \Omega \to [0, 1]$ is the observation function defining the probability of observing $o$ after taking action $a$ and reaching state $s$; $\gamma \in [0, 1)$ is the discount factor determining the importance of future rewards. The objective is to find a policy $\pi$ that maximizes the expected discounted cumulative reward: $\mathbb{E}_\pi \left[ \sum_{t=0}^{\infty} \gamma^t R(s_t, a_t) \right]$, where $a_t \sim \pi(\cdot|o_{1:t})$ depends on the history of observations rather than the true state. Relying on partial observations makes POMDPs harder to solve than MDPs.

### 3.2 MEMORY-INTENSIVE ENVIRONMENTS

Memory-intensive environment is an environment where agents must leverage past experiences to make decisions, often in problems with long-term dependencies or delayed rewards. More formally, following Cherepanov et al. (2024a), a memory-intensive task is a POMDP where there exists a correlation horizon $\xi > 1$, representing the minimum number of timesteps between an event critical for decision-making and when that information must be recalled. Popular memory-intensive environments in RL are listed in Table 2. One way to solving memory-intensive environments is to augment agents with memory mechanisms (see Appendix F).

### 3.3 ROBOTIC TABLETOP MANIPULATION

Robotic tabletop manipulation (Shridhar et al., 2022) involves robots manipulating objects on flat surfaces through actions like grasping, pushing, and picking. While crucial for real-world applications (Levine et al., 2018), most existing simulators treat these tasks as MDPs without memory requirements, failing to capture the spatio-temporal dependencies present in real scenarios. This limitation hinders the development of memory-enhanced agents for practical applications.

## 4 CLASSIFICATION OF MEMORY-INTENSIVE TASKS

The evaluation of memory capabilities in RL faces two major challenges. First, as shown in Table 2, research studies use different sets of environments with minimal overlap, making it difficult to compare memory-enhanced agents across studies. Second, even within individual studies, benchmarks may focus on testing similar memory aspects (e.g., remembering object locations) while neglecting others (e.g., reconstructing sequential events), leading to incomplete evaluation of agents' memory.

Different architectures may exhibit varying performance across memory tasks. For instance, an architecture optimized for long-term object property recall might struggle with sequential memory tasks, yet these limitations often remain undetected due to the narrow focus of existing evaluation approaches.

Figure 3: MIKASA bridges the gap between human-like memory complexity and RL agents requirements. While agents tasks don't require the full spectrum of human memory capabilities, they can't be reduced to simple spatio-temporal dependencies. MIKASA provides a balanced framework that captures essential memory aspects for agents tasks while maintaining practical simplicity.

To address these challenges, we propose a systematic approach to memory evaluation in RL. Drawing from established research in developmental psychology and cognitive science, where similar memory challenges have been extensively studied in humans, we develop a categorization framework consisting of four distinct memory task classes, detailed in Section 4.2.

## 4.1 MEMORY: FROM COGNITIVE SCIENCE TO RL

In developmental psychology and cognitive science, memory is classified into categories based on cognitive processes. Key concepts include object permanence (Piaget, 1952), which involves remembering the existence of objects out of sight, and categorical perception (Liberman et al., 1957), where objects are grouped based on attributes like color or shape. Working memory (Baddeley, 1992) and memory span (Daneman & Carpenter, 1980) refer to the ability to hold and manipulate information over time, while causal reasoning (Kuhn, 2012) and transitive inference (Heckers et al., 2004) involve understanding cause-and-effect relationships and deducing hidden relationships, respectively.

The RL field has attempted to utilize these concepts in the design of specific memory-intensive environments Fortunato et al. (2020); Lampinen et al. (2021), but these have been limited at the task design level. Of particular interest, however, is how existing memory-intensive tasks can be categorized using these concepts to develop a benchmark on which to test the greatest number of memory capabilities of memory-enhanced agents, and it is this problem that we address in this paper. Thus, we aim to provide a balanced framework that covers important aspects of memory for real-world applications while maintaining practical simplicity (see Figure 3).

## 4.2 TAXONOMY OF MEMORY TASKS

We introduce a comprehensive task classification framework for evaluating memory mechanisms in RL. Our framework categorizes memory-intensive tasks into four fundamental types, each targeting distinct aspects of memory capabilities:

1. **Object Memory.** Tasks that evaluate an agent's ability to maintain object-related information over time, particularly when objects become temporarily unobservable. These tasks align with the cognitive concept of object permanence, requiring agents to track object properties when occluded, maintain object state representations, and recognize encountered objects. Example: a robot remembers which fruit it put in the fridge.

2. **Spatial Memory.** Tasks focused on environmental awareness and navigation, where agents must remember object locations, maintain mental maps of environment layouts, and navigate based on previously observed spatial information. Example: the robot remembers the position of a mug it moved while cleaning and returns it to its place.

3. **Sequential Memory.** Tasks that test an agent's ability to process and utilize temporally ordered information, similar to human serial recall and working memory. These tasks require remembering action sequences, maintaining order-dependent information, and using past decisions to inform future actions. Example: a robot memorizes the order of the ingredients it has added to a soup.

4. **Memory Capacity.** Tasks that challenge an agent's ability to manage multiple pieces of information simultaneously, analogous to human memory span. These tasks evaluate information retention limits and multi-task information processing. Example: a robot is able to memorize the positions of several different objects while cleaning a table.

This classification framework enables systematic evaluation of memory-enhanced RL agents across diverse scenarios. By providing a structured approach to memory task categorization, we establish a

foundation for comprehensive benchmarking that spans the wide spectrum of memory requirements. In the following section, we present a carefully curated set of tasks based on this classification, forming the basis of our proposed MIKASA benchmark.

## 5 MIKASA-BASE

**Motivation and Overview.** Despite the importance of memory in decision-making, the RL community lacks standardized tools for benchmarking memory capabilities. Existing studies typically introduce bespoke environments tailored to their proposed algorithms, leading to fragmentation and limited comparability across works (see Table 2). Moreover, many popular memory benchmarks focus narrowly on specific memory types, overlooking the diversity of memory demands found in real-world applications. To address this gap, we introduce **MIKASA-Base**, a unified benchmark that consolidates widely used open-source memory-intensive environments under a common Gym-like API. Our goal is to streamline reproducibility, support fair comparisons, and promote systematic evaluation of memory in RL.

Table 3: Analysis of established robotics frameworks with manipulation tasks, comparing their support for memory-intensive tasks. † – excluding Franka Kitchen. * – concurrent work with three memory tasks with only one type of memory.

| Robotics Framework with Manipulation Tasks | Memory Tasks | | |
|---|---|---|---|
| | Manipulation | Atomic | Low-level actions |
| MIKASA-Robo (**Ours**) | ✓ | ✓ | ✓ |
| MemoryBench* (Fang et al., 2025) | ✓ | ✓ | ✓ |
| ManiSkill3 (Tao et al., 2024) | ✗ | ✗ | ✗ |
| ManiSkill-HAB (Shukla et al., 2024) | ✗ | ✗ | ✗ |
| FetchBench (Han et al., 2024) | ✗ | ✗ | ✗ |
| RoboCasa (Nasiriany et al., 2024) | ✗ | ✗ | ✗ |
| Gymnasium-Robotics† (de Lazcano et al., 2024) | ✗ | ✗ | ✗ |
| BEHAVIOR-1K (Li et al., 2024) | ✓ | ✗ | ✗ |
| LIBERO (Liu et al., 2023) | ✓ | ✗ | ✗ |
| ARNOLD (Gong et al., 2023) | ✗ | ✗ | ✗ |
| LoHoRavens (Zhang et al., 2023) | ✗ | ✗ | ✗ |
| iGibson 2.0 (Li et al., 2022) | ✓ | ✗ | ✗ |
| VIMA (Jiang et al., 2022) | ✓ | ✓ | ✗ |
| Isaac Sim (Makoviychuk et al., 2021) | ✗ | ✗ | ✗ |
| panda-gym (Gallouédec et al., 2021) | ✗ | ✗ | ✗ |
| Ravens (Zeng et al., 2021) | ✗ | ✗ | ✗ |
| Habitat 2.0 (Szot et al., 2021) | ✗ | ✗ | ✗ |
| Meta-World (Yu et al., 2020) | ✗ | ✗ | ✗ |
| CausalWorld (Ahmed et al., 2020) | ✗ | ✗ | ✗ |
| RLBench (James et al., 2020) | ✗ | ✗ | ✗ |
| robosuite (Zhu et al., 2020b) | ✗ | ✗ | ✗ |
| dm_control (Tunyasuvunakool et al., 2020) | ✗ | ✗ | ✗ |
| Franka Kitchen (Gupta et al., 2019) | ✗ | ✗ | ✗ |
| SURREAL (Fan et al., 2018) | ✗ | ✗ | ✗ |
| AI2-THOR (Kolve et al., 2017) | ✗ | ✗ | ✗ |

reproducibility, support fair comparisons, and promote systematic evaluation of memory in RL.

**Benchmark Design Principles.** MIKASA-Base is designed around core principles that support rigorous and interpretable evaluation of memory in RL. To disentangle memory from unrelated challenges, we organize tasks into two tiers. The first tier consists of **diagnostic** vector-based environments that isolate specific memory mechanisms. The second tier includes **complex** image-based tasks that incorporate realistic perception challenges, thus more closely resembling real-world settings. This hierarchical structure enables researchers to validate memory capabilities incrementally – from atomic reasoning to high-dimensional sensory input.

**Task Classification and Selection.** Building on our taxonomy from Section 4.2, we systematically reviewed open-source memory benchmarks and categorized their tasks into four distinct types of memory usage. We selected a diverse yet representative subset of environments to cover this taxonomy – ranging from object permanence to sequential planning. All selected tasks are unified under a single, consistent API. Descriptions are provided in Appendix L, and an overview of MIKASA-Base tasks appears in Table 9. This consolidation supports architectural ablations, direct comparison of methods, and simplified evaluation pipelines. Implementation details can be found in Appendix E.

MIKASA-Base provides the first systematic and unified benchmark for evaluating memory in RL. It mitigates fragmentation by standardizing task access and evaluation, and its structured progression enables precise attribution of memory-related agent failures. By covering a broad spectrum of memory challenges within a common framework, MIKASA-Base offers a foundation for robust, reproducible research in memory-centric RL.

## 6 MIKASA-ROBO

The landscape of robotic manipulation frameworks reveals significant limitations in addressing memory-intensive tasks. While partial observability is well-studied in navigation, manipulation

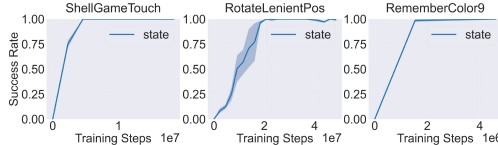

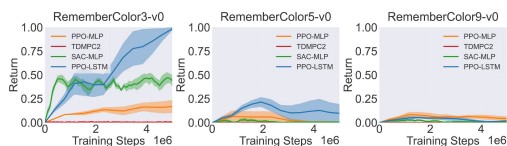

Figure 4: Performance of PPO-MLP trained in `state` mode, i.e., in MDP mode without the need for memory. These results suggest that the proposed tasks are inherently solvable with a success rate of 100%.

Figure 5: Online RL baselines with MLP and LSTM backbones trained in `RGB+joints` mode on the `RememberColor-v0` environment with dense rewards. Both architectures fail to solve medium and high complexity tasks.

scenarios are still predominantly evaluated under full observability, with limited focus on memory demands (see Table 3). Among frameworks that do consider memory, BEHAVIOR-1k (Li et al., 2024) and iGibson 2.0 (Li et al., 2022) include highly complex, non-atomic tasks, which obscure the evaluation of specific memory mechanisms. VIMA (Jiang et al., 2022) relies on high-level action abstractions, limiting temporal memory assessment. To address these gaps, we introduce **MIKASA-Robo**, a benchmark specifically designed to evaluate diverse memory skills in robotic manipulation through well-isolated, fine-grained tasks.

Concurrently with our work, Fang et al. (2025) proposed **MemoryBench**, a benchmark focused on spatial memory with three robotic tasks. In contrast, MIKASA-Robo spans four memory categories and 32 tasks, enabling broader and more systematic evaluation of memory mechanisms in RL agents.

**MIKASA-Robo** is a benchmark designed for memory-intensive robotic tabletop manipulation tasks, simulating real-world challenges commonly encountered by robots. These tasks include locating occluded objects, recalling previous configurations, and executing complex sequences of actions over extended time horizons. By incorporating meaningful partial observability, this framework offers a systematic approach to test an agent's memory mechanisms.

Building upon the robust foundation of ManiSkill3 framework (Tao et al., 2024), our benchmark leverages its efficient parallel GPU-based training capabilities to create and evaluate these tasks.

## 6.1 MIKASA-ROBO MANIFESTATION

In designing the tasks, we drew inspiration from the four memory types identified in our classification framework (Section 4.2). We developed **32 tasks across 12 categories of robotic tabletop manipulation**, each targeting specific aspects of object memory, spatial memory, sequential memory, and memory capacity. These tasks feature varying levels of complexity, allowing for systematic evaluation of different memory mechanisms. For instance, some tasks test object permanence by requiring the agent to track occluded objects, while others challenge sequential memory by requiring the reproduction of a strict order of actions. A summary of these tasks and their corresponding memory types is provided in Table 1, with detailed descriptions in Appendix K. Information on task customization, including adjustments of time horizons and environment parameters, can be found in Appendix M.

To illustrate the concept of our memory-intensive framework, we present `ShellGameTouch-v0`, `RememberColor-v0`, and `RotateLenientPos-v0` tasks in Figure 2. In the `ShellGameTouch-v0` task, the agent observes a red ball placed in one of three positions over the first 5 steps ($t \in [0, 4]$). At $t = 5$, the ball and the three positions are covered by mugs. The agent must then determine the location of the ball by interacting with the correct mug. In the simplest mode (`Touch`), the agent only needs to touch the correct mug, whereas in other modes, it must either push or lift the mug. In the `RememberColor-v0` task, the agent observes a cube of a specific color for 5 steps ($t \in [0, 4]$). After the cube disappears for 5 steps, 3, 5, or 9 (depending on task mode) cubes of different colors appear at $t = 10$. The agent's task is to identify and select the same cube it initially saw. In the `RotateLenientPos-v0` task, the agent must rotate a randomly oriented peg by a specified clockwise angle.

The MIKASA-Robo benchmark offers multiple training modes: `state` (complete vector information including oracle data and Tool Center Point (TCP) pose), `RGB` (top-view and gripper-camera images with TCP position), `joints` (joint states and TCP pose), `oracle` (task-specific environment data

for debugging), and `prompt` (static task instructions). While any mode combination is possible, **RGB+joints serves as the standard memory testing configuration**, with `state` mode reserved for MDP-based tasks.

The MIKASA-Robo benchmark implements two types of reward functions: dense and sparse. The dense reward provides continuous feedback based on the agent's progress towards the goal, while the sparse reward only signals task completion. While dense rewards facilitate faster learning in our experiments, sparse rewards better reflect real-world scenarios where intermediate feedback is often unavailable, making them crucial for evaluating practical applicability of memory-enhanced agents.

## 6.2 ONLINE RL BASELINES

For the experimental evaluation, we chose on-policy Proximal Policy Optimization (PPO, (Schulman et al., 2017)) with two underlying architectures: Multilayer Perceptron (MLP) and Long Short-Term Memory (LSTM, (Hochreiter & Schmidhuber, 1997)), as well as popular in robotics off-policy Soft Actor-Critic (SAC, (Haarnoja et al., 2018)) and model-based Temporal Difference Learning for Model Predictive Control (TD-MPC2, (Hansen et al., 2024)).

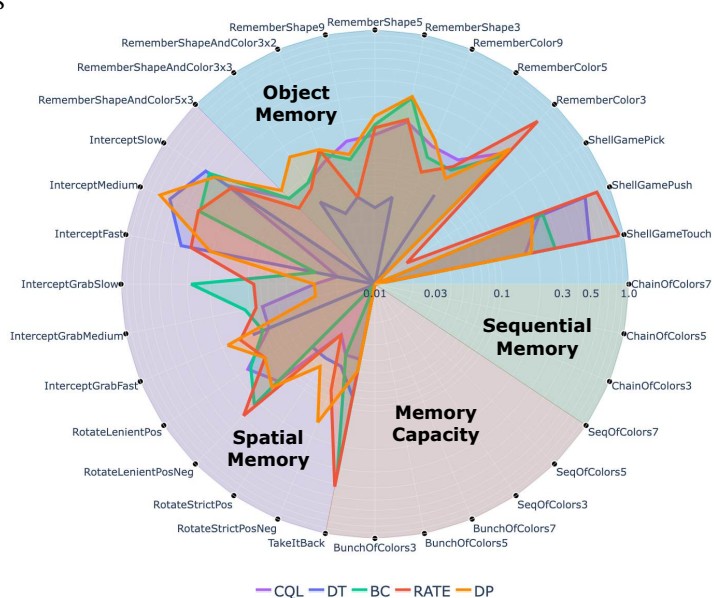

Figure 6: Results of Offline RL baselines with memory (RATE, DT) and without memory (BC-MLP, CQL-MLP, DP) on all 32 MIKASA-Robo tasks. Training was performed in `RGB` mode with sparse rewards (success condition).

The MLP variant serves as a memory-less baseline, while LSTM represents a widely-adopted memory mechanism in RL, known for its effectiveness in solving POMDPs (Ni et al., 2021). This choice of architectures enables direct comparison between memory-less and memory-enhanced agents while validating our benchmark's ability to assess memory. We focus specifically on these fundamental architectures as they align with our primary goal of benchmark validation rather than comprehensive algorithm comparison. To demonstrate that all proposed environments are solvable with 100% success rate (SR), we trained a PPO-MLP agent using `state` mode, where it had full access to system information. Results for select tasks are shown in Figure 4; full results are in Appendix G.

Training under the `RGB+joints` mode with dense rewards reveals the memory-intensive nature of our tasks. Using the `RememberColor-v0` task as an example, PPO-LSTM demonstrates superior performance compared to PPO-MLP when distinguishing between three colors (see Figure 5). However, both agents' success rates drop dramatically to near-zero as the task complexity increases to five or nine colors. Moreover, under sparse reward conditions, both architectures fail to solve even the three-color variant (see Appendix G, Figure 13). Additionally, our findings indicate that, while SAC and TD-MPC2 exhibit higher sample efficiency compared to PPO-MLP, when faced with more complex challenges, the lack of an explicit memory mechanism becomes a critical shortcoming, resulting in low performance, which also emphasizes the inappropriateness of algorithms common in the robotics community for memory-intensive tasks. These results validate our benchmark's effectiveness in evaluating agents' memory, showing clear performance degradation as memory demands increase.

Table 4: Performance of VLA models on selected memory-intensive tasks from the MIKASA-Robo benchmark. Reported values denote average success rates over 100 evaluation episodes (mean ± sem). Tasks include spatial reasoning (`ShellGameTouch`, `InterceptMedium`) and color-based memory retrieval (`RememberColor3/5/9`).

| Model | ShellGameTouch | InterceptMedium | RememberColor3 | RememberColor5 | RememberColor9 |
|---|---|---|---|---|---|
| Octo-small | $0.46 \pm 0.05$ | $0.39 \pm 0.04$ | $0.45 \pm 0.06$ | $0.17 \pm 0.03$ | $0.11 \pm 0.03$ |
| OpenVLA ($K=4$) | $0.12 \pm 0.05$ | $0.06 \pm 0.02$ | $0.21 \pm 0.00$ | $0.09 \pm 0.02$ | $0.08 \pm 0.02$ |
| OpenVLA ($K=8$) | $0.47 \pm 0.05$ | $0.14 \pm 0.03$ | $0.59 \pm 0.04$ | $0.16 \pm 0.03$ | $0.06 \pm 0.02$ |
| SpatialVLA ($K=4$) | $0.23 \pm 0.04$ | $0.27 \pm 0.04$ | $0.27 \pm 0.05$ | $0.17 \pm 0.03$ | $0.11 \pm 0.03$ |
| $\pi_0$ ($K=4$) | $0.33 \pm 0.05$ | $0.42 \pm 0.03$ | $0.35 \pm 0.04$ | $0.22 \pm 0.04$ | $0.15 \pm 0.02$ |

## 6.3 OFFLINE RL BASELINES

Since dense rewards are typically not available in the real world, it is of particular interest to train on sparse rewards represented as a binary flag of a successfully completed episode. Whereas models with online learning are extremely hard to handle in this setting, we also conducted experiments with five Offline RL models: Decision Transformer (DT) (Chen et al., 2021)) and Recurrent Action Transformer with Memory (RATE) (Cherepanov et al., 2024b)) based on the Transformer architecture, Standard Behavioral Cloning (BC) and Conservative Q-Learning (CQL) (Kumar et al., 2020)) with MLP backbones, as well as Diffusion Policy (DP) (Chi et al., 2023)) – a recent and popular approach in robotic manipulation that leverages diffusion models for direct action prediction.

Experimental results with Offline RL models trained using two RGB camera views and sparse rewards are presented in Figure 6. As can be seen from Figure 6, none of the models – including those explicitly designed for sequence modeling – were able to successfully solve the majority of MIKASA-Robo tasks, demonstrating the challenge posed by the benchmark. Training was performed using datasets consisting of 1000 successful trajectories per task, obtained by using PPO with oracle-level information about the environment (see details in Appendix C).

Notably, none of the evaluated models were able to solve tasks requiring high Memory Capacity or Sequential Memory, further underscoring their complexity. More detailed results for Offline RL algorithms are presented in Appendix, Table 8.

## 6.4 VLA BASELINES

To investigate the capabilities of state-of-the-art Visual-Language-Action (VLA) models in memory-intensive robotic tasks, we selected four representative baselines: Octo (Team et al., 2024), Open-VLA (Kim et al., 2024), $\pi_0$ (Black et al., 2024), and SpatialVLA (Qu et al., 2025). Although none claims to implement sophisticated memory mechanisms, the experiments offer insights into existing memory capabilities in VLA models.

Octo is a transformer with diffusion heads pretrained on Open X-Embodiment (Collaboration et al., 2025); we fine-tuned only the readout heads, using the full pretrained context length of 10 and action chunk size (K=4). OpenVLA uses a Prismatic-7B backbone (Karamcheti et al., 2024), fine-tuned with LoRA adapters, chunking, and $L_1$ loss (Kim et al., 2025). $\pi_0$ combines a pretrained VLM with a lightweight flow-matching expert. SpatialVLA augments a VLA with egocentric 3D position encodings and discretized action grids. We evaluate chunk sizes K=4 and K=8. All models were trained on 250 expert trajectories per task, using $128 \times 128$ RGB image pairs (base and wrist views) and end-effector control (see Appendix D).

Experimental results (Table 4) reveal notable trends. Octo (context size 10) outperforms random on simpler tasks, suggesting some innate memory capacity, but degrades with complexity, indicating limited scalability. OpenVLA behaves differently across chunk sizes: with $K = 8$, it exceeds random on tasks like `RememberColor3` and `ShellGameTouch`, despite lacking step-wise history. However, performance drops on harder tasks. With $K = 4$, OpenVLA, SpatialVLA, and $\pi_0$ fail across the board, showing random-like performance. These results suggest larger chunks can bypass memory by generating full trajectories from early cues, but this fails with smaller chunks, where initially correct actions often collapse into confusion. Thus, chunking offers limited compensation for lack of memory. We also conducted real-world experiments using the $\pi_{0.5}$ model (Black et al., 2025) (see Appendix N), where the empirical evidence further corroborates the central conclusion that modern VLA models lack the ability to retain task-relevant information over long temporal horizons. The results reproduce the same pattern observed in simulation, with strong performance on fully observable tasks and a sharp degradation once the task introduces an occluded interval that requires persistent memory.

The sharp decline on harder tasks underscores the need for dedicated memory architectures and validates the multi-difficulty hierarchy in MIKASA-Robo to prevent such "shortcuts." Our experiments with Octo, OpenVLA, $\pi_0$, and SpatialVLA highlight a critical gap in current VLA models: without effective long-term memory, performance is brittle on tasks requiring strong memory. These findings reveal current limitations and reinforce the value of the MIKASA-Robo benchmark.

## 7  LIMITATIONS AND FUTURE WORK

Future work could explore more extensive adaptation of large VLA models within MIKASA to obtain a clearer picture of their memory capabilities. A complementary direction is to broaden the benchmark to encompass memory phenomena that fall outside the current focus on spatio-temporal dependencies within single episodes. The present tasks do not capture higher-level processes such as meta-RL. These settings require problem formulations that extend beyond the POMDP structure instantiated in the benchmark, since they involve reasoning over distributions of tasks rather than isolated trajectories. It would also be valuable to study how agents cope with spurious correlations, interference between stored representations, and the dynamics of memory overwriting or forgetting under limited memory capacity, all of which remain unexplored in the current framework. Finally, developing of additional evaluation metrics, instead of relying solely on success rates or returns, offers a promising path toward more precise and discriminative assessment of memory mechanisms.

## 8  CONCLUSION

We present **MIKASA**, a unified benchmark suite for evaluating memory in RL. Our work addresses key gaps in the field by introducing: (1) a taxonomy of memory types, spanning object, spatial, sequential, and capacity requirements; (2) **MIKASA-Base**, a standardized collection of open-source memory tasks; (3) **MIKASA-Robo**, a suite of 32 robotic manipulation tasks designed to isolate and stress specific forms of memory; and (4) accompanying offline datasets to support reproducible large-scale evaluation. Experiments with online, offline, and VLA agents consistently show that existing methods struggle when memory requirements become substantial, which highlights the need for architectures with explicit and robust long-horizon memory mechanisms. We further validated our findings through real-world experiments on a physical robot platform. The results mirror the simulation hierarchy, with reliable performance in fully observable tasks, partial degradation under dynamic but non-occluded conditions, and failure when long-horizon occlusion is introduced. This consistency confirms that memory, rather than embodiment or perception noise, is the primary limitation. MIKASA is intended to guide and accelerate progress in memory-intensive RL for real-world applications. The **MIKASA-Robo** suite is open-source under the MIT license and can be conveniently installed via `pip install mikasa-robo-suite`.

### ACKNOWLEDGEMENTS
We thank Nuraddin Kerimov for providing the SO-101 robotic arm and for his extensive assistance with its setup and the execution of the real-world experiments. We also thank Andrei Spiridonov for his help with the robot and for valuable technical consultations during the course of this work.

### REPRODUCIBILITY STATEMENT
We have taken several measures to ensure the reproducibility of our work. All benchmark tasks, including **MIKASA-Base** and **MIKASA-Robo**, are publicly released under the MIT license with complete code and installation instructions (see Appendix E and Appendix A) (we will publish the datasets after the rebuttal period). Detailed environment descriptions and task customization guides are provided in Appendix K and Appendix M. For experiments, we report training and evaluation protocols in Appendix I, including random seeds, number of runs, and hardware setup. Offline datasets used in our experiments are released in processed form together with scripts for collection and preprocessing (see Appendix C). Architectural and algorithmic details for all online, offline, and VLA baselines are described in Section 6.2, Section 6.3, and Section 6.4. All additional implementation details and hyperparameters are provided in the Appendix. An anonymous repository containing the source code, dataset download links, and pretrained models will be submitted with the supplementary materials to facilitate full reproducibility of our results.

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

# Table of Contents

## A  MIKASA-ROBO IMPLEMENTATION DETAILS

An example of running the environment from the MIKASA-Robo benchmark is shown in Code 1. For ease of debugging, we also added various wrappers (found in `mikasa_robo_suite/utils/wrappers/`) that display useful information about the episode on the video (Code 2). Thus, `RenderStepInfoWrapper()` displays the current step in the environment; `DebugRewardWrapper()` displays information about the full reward at the current step in the environment; `DebugRewardWrapper()` displays information about each component that generates the reward function at the current step. In addition, we also added task-specific wrappers for each environment. For example, `RememberColorInfoWrapper()` displays the target color of the cube in the `RememberColor-v0` task, and `ShellGameRenderCupInfoWrapper()` displays which mug the ball is actually under in the `ShellGame-v0` task.

Code 1: Getting started with MIKASA-Robo using the `RememberColor9-v0` environment.

```python
# pip install mikasa-robo-suite
import mikasa_robo_suite
from mikasa_robo_suite.utils.wrappers import
    ↪ StateOnlyTensorToDictWrapper
from tqdm.notebook import tqdm
import torch
import gymnasium as gym

# Create the environment via gym.make()
# obs_mode="rgb" for modes "RGB", "RGB+joint", "RGB+oracle" etc.
# obs_mode="state" for mode "state"
episode_timeout = 90
env = gym.make("RememberColor9-v0", num_envs=512 obs_mode="rgb",
    ↪ render_mode="all")
env = StateOnlyTensorToDictWrapper(env) # * always use this wrapper!

obs, _ = env.reset(seed=42)
print(obs.keys())
for i in tqdm(range(episode_timeout)):
    action = torch.from_numpy(env.action_space.sample())
    obs, reward, terminated, truncated, info = env.step(action)

env.close()
```

Code 2: MIKASA-Robo wrappers system.

```python
import mikasa_robo_suite, torch
from mikasa_robo_suite.dataset_collectors.get_mikasa_robo_datasets
    ↪ import env_info
import gymnasium as gym
from mani_skill.utils.wrappers import RecordEpisode
from IPython.display import Video

env = gym.make("RememberColor9-v0", num_envs=512, obs_mode="rgb",
    ↪ render_mode="all")
state_wrappers_list, episode_timeout = env_info("RememberColor9-v0")
for wrapper_class, wrapper_kwargs in state_wrappers_list:
    env = wrapper_class(env, **wrapper_kwargs)
env = RecordEpisode(env, f"./videos", max_steps_per_video=
    ↪ episode_timeout)

obs, _ = env.reset(seed=42)
for i in range(episode_timeout):
    action = torch.from_numpy(env.action_space.sample())
    obs, reward, terminated, truncated, info = env.step(action)

Video(f"./videos/0.mp4", embed=True, width=640)
env.close()
```

## B  MIKASA-ROBO TASKS CUSTOMIZATION

Beyond the official configurations, **MIKASA-Robo** is designed to be fully customizable. Researchers can directly adjust environment parameters – such as number of objects, episode length, cue duration, or delay intervals – to create variants tailored for debugging, ablation studies, or curriculum learning. This flexibility ensures that tasks can scale from minimal examples to extremely challenging settings while preserving their memory-centric nature.

We deliberately include highly challenging variants (e.g., `BunchOfColors`, `SeqOfColors`, `ChainOfColors`) to ensure that the benchmark continues to stress-test algorithms as memory capabilities advance. To avoid hidden shortcuts, we provide a set of fixed *official configurations* with multiple difficulty levels (e.g., `RememberColor3/5/9`). At the same time, each environment exposes its underlying parameters, making customization straightforward.

For example, the `RememberColor` task can be customized as follows:

Code 3: Customized `RememberColor` environment.

```python
from mani_skill.utils.registration import register_env
from mikasa_robo_suite.remember_color import RememberColorBaseEnv
import gymnasium as gym

@register_env("RememberColor4Debug-v0", max_episode_steps=1000)
class RememberColorDebugEnv(RememberColorBaseEnv):
    COLORS       = 4     # 1-9 unique cubes
    TIME_OFFSET  = 200   # duration target cube is visible
    GOAL_THRESH  = 0.03  # success threshold
    CUBE_HALFSIZE= 0.02  # cube size
    DELTA_TIME   = 100   # delay before response

env = gym.make("RememberColor4Debug-v0", num_envs=256,
               obs_mode="rgb", render_mode="all",
               delta_time=DELTA_TIME)
```

Similarly, the `SeqOfColors` task can be configured with custom sequence length and timing:

Code 4: Customized `SeqOfColors` environment.

```python
from mani_skill.utils.registration import register_env
from mikasa_robo_suite.seq_of_colors import SeqOfColorsEnv
import gymnasium as gym

@register_env("SeqOfColors6Debug-v0", max_episode_steps=1000)
class SeqOfColorsDebugEnv(SeqOfColorsEnv):
    COLORS          = 2   # 1-9 unique cubes
    GOAL_THRESH     = 0.03
    CUBE_HALFSIZE   = 0.02
    SEQUENCE_LENGTH = 2   # number of cubes in sequence
    STEP_DURATION   = 15  # duration per cube
    EMPTY_DURATION  = 5   # empty delay between cubes

env = gym.make("SeqOfColors6Debug-v0", num_envs=256,
               obs_mode="rgb", render_mode="all")
```

The design of **MIKASA-Robo** emphasizes isolating the role of memory, rather than long-horizon credit assignment. For example, tasks like `RememberColor` already become non-Markovian after a single occlusion, so even short horizons (e.g., 60 steps) suffice to reveal memory limitations. Still, researchers can easily scale horizon length and difficulty by tuning memory-related parameters.

Below we illustrate how to extend `RememberColor` into a long-horizon setting, increasing both episode length and delay:

Code 5: Long-horizon variant of `RememberColor`.

```python
from mani_skill.utils.registration import register_env
from mikasa_robo_suite.remember_color import RememberColorBaseEnv
import gymnasium as gym

@register_env("RememberColor3Debug-v0", max_episode_steps=1000)
class RememberColorDebugEnv(RememberColorBaseEnv):
    COLORS       = 3
    TIME_OFFSET  = 50    # target cube duration
    GOAL_THRESH  = 0.03
    CUBE_HALFSIZE= 0.02
    DELTA_TIME   = 900   # extended delay

env = gym.make("RememberColor3Debug-v0", num_envs=256,
               obs_mode="rgb", render_mode="all",
               delta_time=DELTA_TIME)
```

This flexibility applies to all tasks in the benchmark, making **MIKASA-Robo** suitable for controlled debugging, systematic ablations, or curriculum-based studies of memory.

**Embodiment flexibility in MIKASA-Robo.** Memory requirements in MIKASA-Robo are independent of the robot embodiment. The embodiment determines kinematics, actuator morphology, and low-level motion feasibility, but the memory load is induced entirely by the task structure and therefore depends on the policy rather than on the hardware platform. We default to the Franka Panda because it is one of the most widely adopted manipulators in RL and robotics research, offering stable physics, high quality simulation assets, and broad community support. However, ManiSkill3 natively supports alternative embodiments, and all MIKASA-Robo tasks can be instantiated with different arms without altering the underlying memory challenge (see Figure 7).

**Extending atomic tasks with realistic object distributions.** The objective of MIKASA-Robo is to isolate the memory dimension of manipulation, and incorporating complex meshes, textures, or fragile grasping conditions introduces confounding factors that mask the memory dependencies we aim to evaluate. Primitive shapes yield clean, well-controlled environments in which performance differences can be attributed to memory mechanisms rather than physics artifacts or mesh-specific interactions. Although the benchmark is designed atomically to isolate memory from perception, control, and other competencies, each task class can be lifted to more realistic regimes. After an agent succeeds on the atomic variants, the same task structures can be re-instantiated using complex objects (for example, from YCB Dataset (Calli et al., 2015) or BridgeData V2 (Walke et al., 2023)), preserving the underlying memory requirements while introducing richer visual variability and more demanding manipulation dynamics. For example, `RememberColor3-v0`, which evaluates recall of the color of one of three cubes, generalizes naturally to `RememberObject3-v0`, where the cubes are replaced with randomly sampled objects whose appearance and geometry necessitate substantially higher perceptual precision and control accuracy (see Figure 8).

**Clutter as an optional difficulty dimension.** We intentionally avoid cluttered scenes in the atomic task suite, since the goal of MIKASA-Robo is to isolate memory from other confounding factors such as occlusion handling, dense object–object interactions, and contact-rich rearrangement. Clutter introduces additional challenges in perception, planning, and control that make it difficult to attribute performance differences to memory mechanisms alone. By keeping the scene minimal, the benchmark ensures that errors arise from failures in memory rather than from the incidental complexity of cluttered manipulation. At the same time, the framework does not restrict users from increasing task difficulty once a policy succeeds on the atomic variants. Clutter can be added in a controlled manner to produce more challenging instances, requiring stronger perception and robustness without altering the underlying memory structure of the original task (see Figure 9).

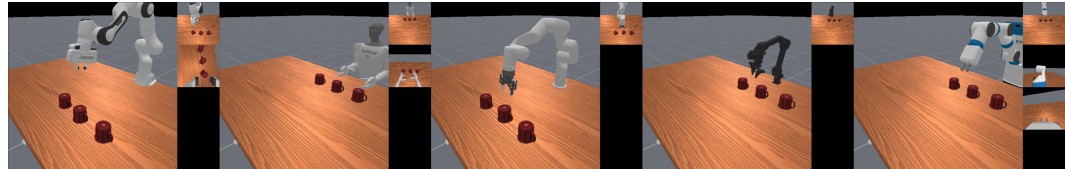

Figure 7: Examples of the `ShellGameTouch-v0` task performed using different robot embodiments available in ManiSkill3. The scenes demonstrate that the benchmark is not tied to a particular manipulator: Franka Panda, Unitree arms, and other platforms can all execute the same task configuration. This illustrates that MIKASA-Robo supports interchangeable embodiments while preserving identical task logic and memory requirements.

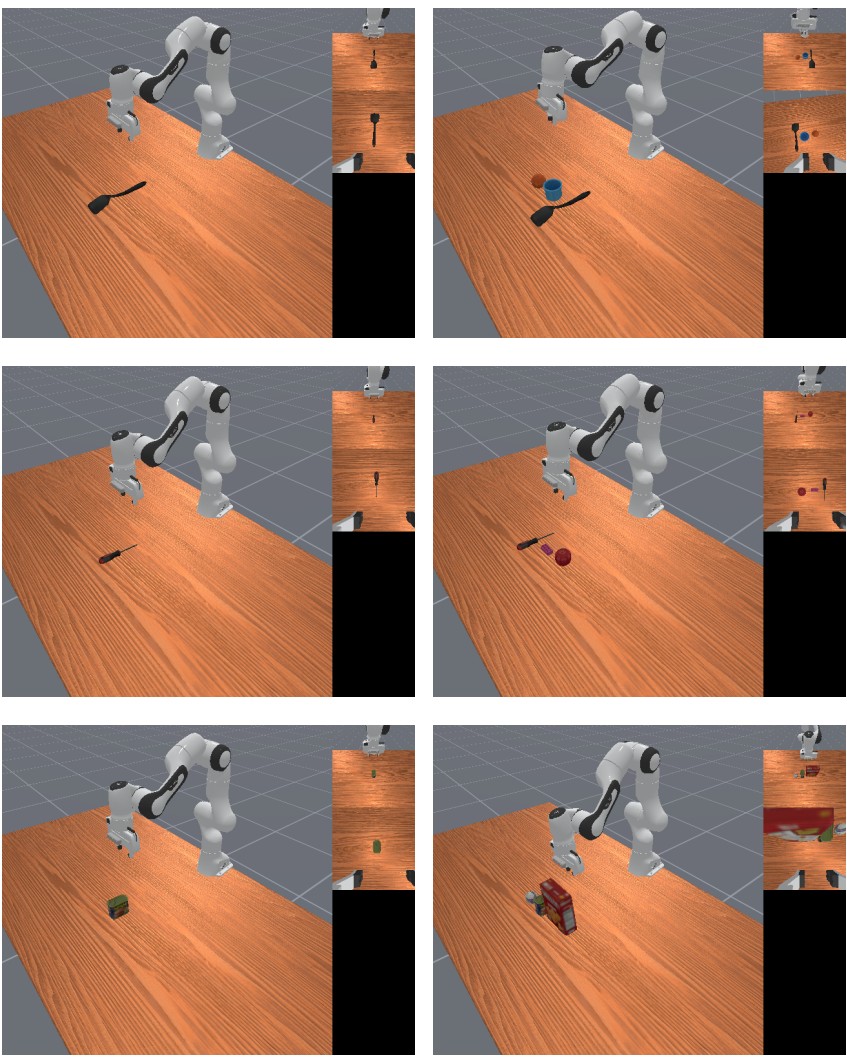

Figure 8: Example rollouts for the `RememberObject3-v0` task. The left column shows the initial state in which the robot observes a single target object. The object is then removed, and the right column shows the subsequent scene populated with distractor objects, at which point the agent must recall and identify the object presented initially. The task preserves the same memory structure as `RememberColor3-v0`, but replaces colored cubes with diverse objects of varying geometry and appearance, thereby increasing perceptual and manipulation complexity while maintaining identical temporal memory requirements.

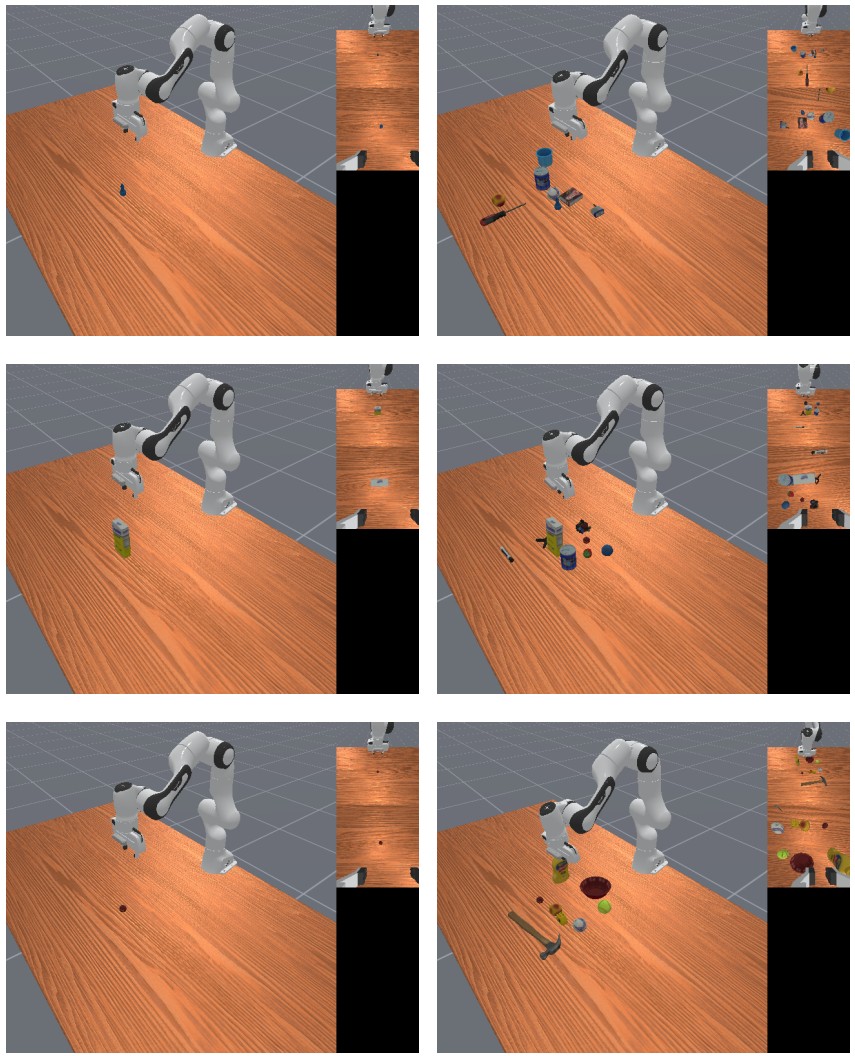

Figure 9: Cluttered variants of the `RememberObject3-v0` task. The left column shows the initial observation containing a single target object, while the right column shows the subsequent scene populated with clutter. Although clutter is not included in the atomic benchmark to avoid confounding perception and contact-rich interactions with memory, it can be added as an optional difficulty dimension once a policy reliably solves the base task. These examples illustrate how clutter increases visual complexity and occlusions.

## C   MIKASA-ROBO DATASETS FOR OFFLINE RL AND IMITATION LEARNING

To train Offline RL or Imitation Learning baselines on camera images (in "RGB" mode) with sparse rewards (success condition), we collected expert-quality datasets for each of the 32 MIKASA-Robo tasks. Datasets were collected using a PPO-MLP agent trained to SR=100% in "state" mode (i.e., with full information about the task being solved) with sparse rewards (success condition). Thus, each dataset is represented by 1000 successful trajectories, where each trajectory consists of:

1. "rgb" (shape: $(T, 128, 128, 6)$) - two RGB images (view from above and from the gripper)
2. "joints" (shape: $(T, 25)$) - Tool Center Point (TCP) position and rotation, and joint positions and velocities
3. "action" (shape: $(T, 8)$) - action (8-dimensional vector)
4. "reward" (shape: $(T, )$) - (dense) reward for each step

Table 5: Tasks configurations for fine-tuning VLA models. The table lists the task ID, number of evaluation steps (T), and the associated language instruction

| Task | T | Language instruction |
|---|---|---|
| `RememberColor3/5/9-v0` | 60 | Remember the color of the cube and then pick the matching one |
| `ShellGameTouch-v0` | 90 | Memorize the position of the cup covering the ball, then pick that cup |
| `InterceptMedium-v0` | 90 | Track the ball's movement, estimate its velocity, then aim the ball at the target |

    5. "success" (shape: $(T,)$) - (sparse) success flag for each step

    6. "done" (shape: $(T,)$) - done flag for each step

These datasets are available for download from the project website. We have also published the weights of the PPO-MLP agent used to collect the dataset, as well as scripts for collecting datasets of any size, to our repository.

## D  MIKASA-ROBO SETUP FOR VLA BASELINES

For experiments involving Vision-Language-Action (VLA) models, we focused on a representative subset of spatial and object memory tasks from MIKASA-Robo. For each task, we generated a dataset of 250 episodes using an oracle PPO policy with full access to the environment state. At every timestep, the policy recorded two synchronized RGB frames (one from the static "base" camera and one from the robot's wrist camera) along with the corresponding end-effector control actions ( `pd_ee_delta_pose` controller from (Tao et al., 2024)). Each task was also paired with a concise language instruction (see Table 5).

All VLA baselines were trained for 50000 iterations and evaluated independently on each task. Complete training/evaluation scripts, language instruction templates, and detailed model hyperparameter settings are provided in the accompanying supplementary code.

## E  MIKASA-BASE IMPLEMENTATION DETAILS

An example of running an environment from the MIKASA-Base benchmark is shown in Code 6. MIKASA-Base supports the standard Gymnasium API and is fully compatible with all its wrappers. This allows users to leverage various functionalities, including parallelization using `AsyncVectorEnv`. MIKASA-Base provides a predefined set of environments with different levels of difficulty. However, users can customize the environment parameters by passing specific arguments (see Code 6).

Code 6: Example code for running `MemoryLength-v0` environment.

```python
import mikasa_base
import gymnasium as gym

# use pre-defined env
# env_id = "MemoryLengthEasy-v0"
# env_kwargs = None

# create env using custom parameters
env_id = "MemoryLength-v0"
env_kwargs = {"memory_length": 10, "num_bits": 1}
seed = 123

env = gym.make(env_id, env_kwargs)

obs, _ = env.reset(seed=seed)

for i in range(11):
    action = env.action_space.sample()
    next_obs, reward, terminations, truncations, infos = env.step(
        ↪ action)
env.close()
```

## F    MEMORY MECHANISMS IN RL

In RL, memory mechanisms are techniques or models used to enable agents to retain and recall information from past interactions with the environment.

There are several approaches to incorporating memory into RL, including recurrent neural networks (RNNs) (Rumelhart et al., 1986; Hochreiter & Schmidhuber, 1997; Chung et al., 2014) which uses hidden states to store information from previous steps (Wierstra et al., 2010; Hausknecht & Stone, 2015), state-space models (SSMs) (Gu et al., 2021; Smith et al., 2023; Gu & Dao, 2023) which uses system state to store historical information (Hafner et al., 2019; Samsami et al., 2024), transformers (Vaswani et al., 2017) which uses attention mechanism to capture sequential dependencies inside the context window (Parisotto et al., 2020; Lampinen et al., 2021; Ni et al., 2023), graph neural networks (GNNs) (Zhou et al., 2020) which uses graphs to store information (Zhu et al., 2023; Kang et al., 2024) etc. Popular agents with memory mechanisms are summarized in Table 2.

Temporal convolutions constitute an additional class of memory mechanisms. Here, information is stored implicitly through convolutional filters applied over the temporal dimension, enabling the agent to integrate multi-step dependencies without explicit recurrent states (YuXuan Liu & Hsieh, 2016; Mishra et al., 2018). Although they lack explicit recurrence or attention, their receptive fields can be designed to capture long-range temporal features. World models (Ha & Schmidhuber, 2018) implement memory by constructing internal predictive dynamics, effectively learning a latent environment model that summarizes prior interactions. External memory mechanisms provide explicit storage that agents can query. Read-only mechanisms such as attention over fixed buffers (Oh et al., 2016; Lampinen et al., 2021; Goyal et al., 2022b; Cherepanov et al., 2024b) allow retrieval without modifying previously stored content, while read-write mechanisms (Graves et al., 2016; Zaremba & Sutskever, 2016; Parisotto & Salakhutdinov, 2017) enable dynamic creation, modification, and deletion of memory entries. These architectures support behaviors analogous to episodic recall or map-building.

Memory can also arise without dedicated architectural components. Some agents encode temporal information directly in their action patterns or latent policies. For instance, temporal intervals or event markers can be encoded implicitly through learned action sequences, allowing the agent to solve memory-intenive tasks despite lacking an explicit memory module. A more recent line of work explores autostigmergy (Deverett et al., 2019), in which memory is externalized through persistent modifications to the environment or to an auxiliary representational medium. Unlike conventional external buffers, autostigmergic mechanisms allow the agent's own interactions to create state traces that influence future decisions, enabling distributed memory without explicit internal storage.

Together, these mechanisms span a wide spectrum, from implicit temporal encoding to explicitly structured memory architectures. They support tasks requiring intra-episode temporal reasoning as well as fast adaptation across tasks. Nonetheless, even when built upon similar foundational architectures, different works operationalize "memory" in distinct ways, leading to heterogeneity in evaluation and interpretation.

## G    CLASSIC BASELINES PERFORMANCE ON THE MIKASA-ROBO BENCHMARK

In this section, we present a comprehensive evaluation of PPO-MLP and PPO-LSTM baselines on our MIKASA-Robo benchmark. Our experiments with PPO-MLP in `state` mode using dense rewards demonstrate perfect performance across all tasks, consistently achieving 100% success rate, as shown in Figure 10 and Figure 11. This remarkable performance serves as a crucial validation of our benchmark design: when an agent has access to complete state information and receives dense rewards, it can master these tasks completely. Therefore, any performance degradation in `RGB+joints` mode observed with other algorithms or training configurations must stem from the algorithmic limitations or learning challenges rather than any inherent flaws in the task design. This empirical evidence confirms that our environments are well-calibrated and properly designed, establishing a solid foundation for evaluating memory-enhanced algorithms. All results are presented as mean $\pm$ standard error of the mean (SEM), where the mean is computed across three independent training runs, and each trained agent is evaluated on 16 different random seeds to ensure robust performance assessment.

The performance evaluation of PPO-MLP and PPO-LSTM with dense rewards in the `RGB+joints` mode is presented in Figure 12. This mode specifically tests the agents' memory capabilities, as it requires remembering and utilizing historical information to solve the tasks. Our results demonstrate a clear distinction between memory-less and memory-enhanced architectures, while also revealing the limitations of conventional memory mechanisms.

Consider the `RememberColor-v0` environment as an illustrative example. In its simplest configuration with three cubes, the memory-less PPO-MLP achieves only 25% success rate. In contrast, PPO-LSTM, leveraging its memory mechanism, achieves perfect performance with 100% success rate. However, as task complexity increases to five or nine cubes, even the LSTM's memory capabilities prove insufficient, with performance degrading significantly.

These results validate two key aspects of our benchmark: first, its effectiveness in distinguishing between memory-less and memory-enhanced architectures, and second, its ability to challenge even sophisticated memory mechanisms as task complexity increases. This demonstrates that MIKASA-Robo provides a competitive yet meaningful evaluation framework for developing and testing advanced memory-enhanced agents.

Our evaluation of PPO-MLP and PPO-LSTM baselines under sparse reward conditions in `RGB+joints` mode reveals the true challenge of our benchmark tasks. As shown in Figure 13, both architectures – even the memory-enhanced LSTM – consistently fail to achieve any meaningful success rate across nearly all considered environments. This striking result underscores the extreme difficulty of memory-intensive manipulation tasks when only terminal rewards are available, highlighting the substantial gap between current algorithms and the level of memory capabilities required for real-world robotic applications.

## H    ADDITIONAL OFFLINE RL RESULTS WITH DENSE REWARDS

In the main paper, we focused on the sparse reward setting for Offline RL, which is particularly challenging since online RL agents are generally ineffective in this regime. To better contextualize the results, we also conducted supplementary experiments in the dense reward setting on a representative subset of tasks: `ShellGameTouch`, `InterceptMedium`, and `RememberColor3/5/9`. We compared four representative algorithms: **RATE**, **DT**, **BC**, and **CQL**.

Table 6: Performance of Offline RL baselines under dense reward formulation.

| Task | RATE | DT | BC | CQL |
|------|------|----|----|-----|
| ShellGameTouch-v0 | 0.97±0.02 | 0.50±0.17 | 0.38±0.03 | 0.02±0.01 |
| InterceptMedium-v0 | 0.39±0.06 | 0.56±0.02 | 0.51±0.03 | 0.04±0.01 |
| RememberColor3-v0 | 0.68±0.04 | 0.05±0.03 | 0.20±0.04 | 0.00±0.00 |
| RememberColor5-v0 | 0.11±0.04 | 0.05±0.03 | 0.13±0.02 | 0.01±0.01 |
| RememberColor9-v0 | 0.10±0.01 | 0.02±0.02 | 0.15±0.03 | 0.01±0.00 |

## I    EXPERIMENTS REPRODUCING AND COMPUTE RESOURCES

All baselines were trained and evaluated under a reproducible standardized setup on a single NVIDIA A100 GPU. For each task, we conducted three independent training runs. Within each run, evaluation was performed over 100 independent episodes with environment and agent random seeds ranging from 1 to 100. We first computed the mean success rate per run, and then report the overall performance as the mean ± standard error (SEM) across the three run-level means.

## J    ADDITIONAL BASELINES: SSMS AND MEMORY-ENHANCED TRANSFORMERS

To further strengthen our evaluation, we included two additional families of baselines in the offline RL setting: (1) **DMamba** (Ota, 2024), a recent state-space model designed for efficient long-sequence

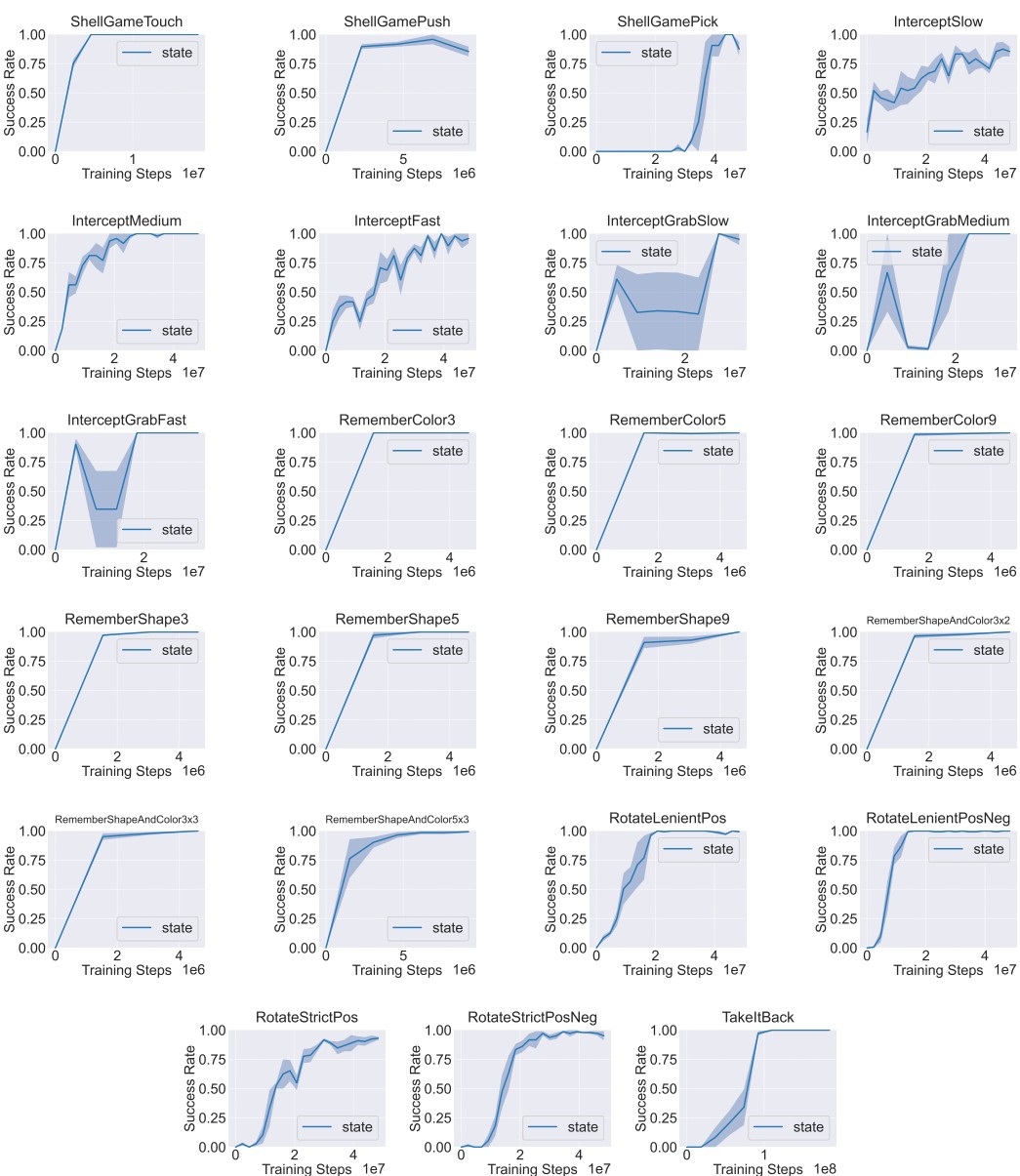

Figure 10: Demonstration of PPO-MLP performance on MIKASA-Robo benchmark when trained with oracle-level `state` information. In this learning mode, MDP problem formulation is considered, i.e. memory is not required for successful problem solving. At the same time, the obtained results show that it is possible to solve these problems and obtain 100% Success Rate.

modeling, and (2) **GTrXL** (Parisotto et al., 2020), a gated recurrent transformer variant proposed specifically, adopted to the offline RL setting.

We tested these methods on a representative subset of tasks – `ShellGameTouch`, `InterceptMedium`, and `RememberColor3/5/9` – and compared them against our primary baselines.

Overall, we find that while GTrXL shows some improvements over standard DT and BC baselines, both it and DMamba still fail to match the performance of RATE model, especially on tasks with higher memory requirements (e.g., `RememberColor5/9`). These results confirm that

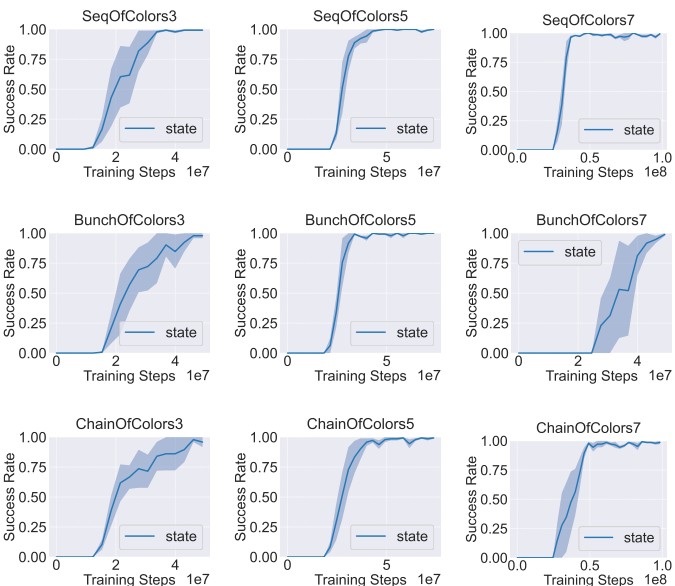

Figure 11: Demonstration of PPO-MLP performance on MIKASA-Robo benchmark when trained with oracle-level `state` information. Results are shown for memory capacity (`SeqOfColors[3,5,7]-v0`, `BunchOfColors[3,5,7]-v0`) and sequential memory (`ChainOfColors[3,5,7]-v0`).

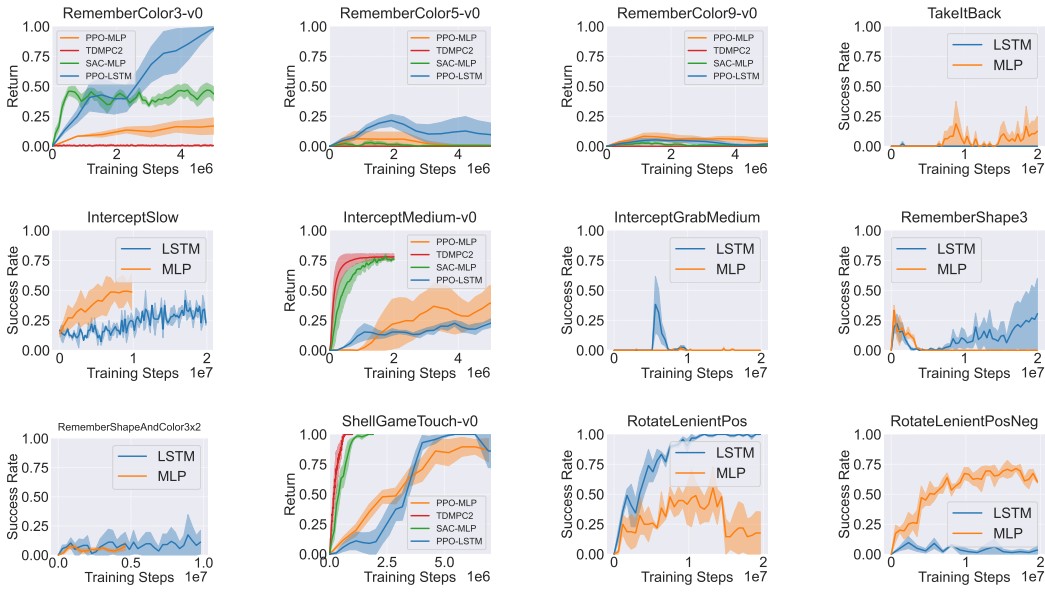

Figure 12: Performance evaluation of PPO-MLP and PPO-LSTM on the MIKASA-Robo benchmark using the "RGB+joints" training mode with dense reward function, where the agent only receives images from the camera (from above and from the gripper) and information about the state of the joints (position and velocity). The results demonstrate that numerous tasks pose significant challenges even for PPO-LSTM agents with memory, establishing these environments as effective benchmarks for evaluating advanced memory-enhanced architectures.

memory-centric SSM and Transformer variants remain challenged by increasing sequence complexity, underscoring the importance of dedicated mechanisms for continual memory retention and rewriting.

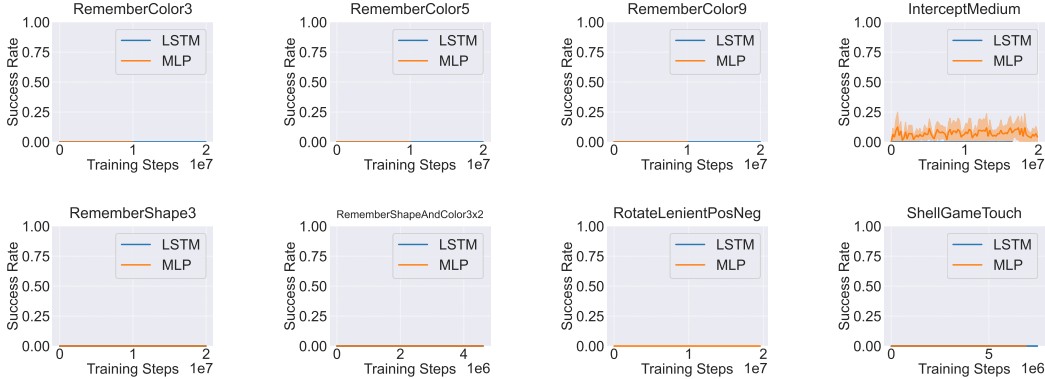

Figure 13: Performance evaluation of PPO-MLP and PPO-LSTM on the MIKASA-Robo benchmark using the "RGB+joints" with sparse reward function training mode, where the agent only receives images from the camera (from above and from the gripper) and information about the state of the joints (position and velocity). This training mode with sparse reward function causes even more difficulty for the agent to learn, making this mode even more challenging for memory-enhanced agents.

Table 7: Offline RL performance of additional SSM/Transformer baselines (DMamba, GTrXL) compared with prior models.

| Task | RATE | DT | BC | CQL | DP | DMamba | GTrXL |
|---|---|---|---|---|---|---|---|
| ShellGameTouch-v0 | 0.92±0.01 | 0.53±0.07 | 0.28±0.01 | 0.16±0.04 | 0.18±0.02 | 0.21±0.02 | 0.80±0.10 |
| InterceptMedium-v0 | 0.09±0.03 | 0.56±0.01 | 0.31±0.14 | 0.03±0.01 | 0.24±0.01 | 0.14±0.07 | 0.64±0.04 |
| RememberColor3-v0 | 0.65±0.04 | 0.01±0.01 | 0.27±0.03 | 0.29±0.01 | 0.07±0.04 | 0.32±0.03 | 0.39±0.06 |
| RememberColor5-v0 | 0.13±0.03 | 0.07±0.05 | 0.12±0.02 | 0.15±0.02 | 0.01±0.01 | 0.11±0.02 | 0.19±0.04 |
| RememberColor9-v0 | 0.09±0.02 | 0.01±0.01 | 0.12±0.02 | 0.15±0.01 | 0.02±0.01 | 0.14±0.00 | 0.16±0.01 |

## K  MIKASA-ROBO DETAILED TASKS DESCRIPTION

In this section, we provide comprehensive descriptions of the 32 memory-intensive tasks that comprise the MIKASA-Robo benchmark. Each task is designed to evaluate specific aspects of memory capabilities in robotic manipulation, ranging from object tracking and spatial memory to sequential decision-making. For each task, we detail its objective, memory requirements, observation space, reward structure, and success criteria. Additionally, we explain how task complexity increases across different variants and discuss the specific memory challenges they present. The following subsections describe each task category and its variants in detail.

Each of the proposed environment supports multiple observation modes:

- **State**: Full state information including ball position
- **RGB+joints**: Two camera views (top-down and gripper) plus robot joint states
- **RGB**: Only visual information from two cameras

In the case of RotateLenient-v0 and RotateStrict-v0, the prompt information available at each step is additionally added to each observation.

Table 8: **Results for Offline RL baselines**. The table shows comparison of transformer-based baselines (RATE, DT), behavior cloning (BC), classic Offline RL baselines (CQL), and Diffusion Policy (DP) on all 32 tasks from the MIKASA-Robo benchmark. Results are presented as mean ± sem across the three runs, where each run is averaged over 100 episodes and sem is the standard error of the mean. Training was performed using only RGB observations (two cameras: top view and gripper view) and using sparse rewards (success once condition). The results show that even models with memory (RATE, DT) are not able to solve most of the benchmark problems, which makes it challenging and promising for further validation of the algorithm.

| # | Environment | RATE | DT | BC | CQL | DP |
|---|---|---|---|---|---|---|
| 1 | ShellGameTouch-v0 | 0.92±0.01 | 0.53±0.07 | 0.28±0.01 | 0.16±0.04 | 0.18±0.02 |
| 2 | ShellGamePush-v0 | 0.78±0.06 | 0.62±0.14 | 0.27±0.01 | 0.25±0.01 | 0.22±0.03 |
| 3 | ShellGamePick-v0 | 0.02±0.01 | 0.00±0.00 | 0.01±0.01 | 0.00±0.00 | 0.01±0.00 |
| 4 | InterceptSlow-v0 | 0.23±0.02 | 0.40±0.02 | 0.37±0.06 | 0.25±0.01 | 0.33±0.05 |
| 5 | InterceptMedium-v0 | 0.32±0.02 | 0.56±0.01 | 0.31±0.14 | 0.03±0.01 | 0.68±0.02 |
| 6 | InterceptFast-v0 | 0.30±0.04 | 0.36±0.04 | 0.03±0.02 | 0.02±0.02 | 0.21±0.05 |
| 7 | InterceptGrabSlow-v0 | 0.09±0.03 | 0.00±0.00 | 0.28±0.18 | 0.03±0.00 | 0.03±0.01 |
| 8 | InterceptGrabMedium-v0 | 0.09±0.03 | 0.00±0.00 | 0.11±0.02 | 0.08±0.04 | 0.03±0.01 |
| 9 | InterceptGrabFast-v0 | 0.14±0.03 | 0.11±0.03 | 0.09±0.02 | 0.08±0.03 | 0.18±0.02 |
| 10 | RotateLenientPos-v0 | 0.11±0.04 | 0.01±0.01 | 0.15±0.03 | 0.16±0.02 | 0.11±0.02 |
| 11 | RotateLenientPosNeg-v0 | 0.29±0.03 | 0.05±0.02 | 0.22±0.01 | 0.12±0.02 | 0.14±0.05 |
| 12 | RotateStrictPos-v0 | 0.03±0.02 | 0.05±0.04 | 0.01±0.00 | 0.03±0.01 | 0.06±0.02 |
| 13 | RotateStrictPosNeg-v0 | 0.08±0.01 | 0.05±0.03 | 0.04±0.02 | 0.04±0.02 | 0.15±0.01 |
| 14 | TakeItBack-v0 | 0.42±0.24 | 0.08±0.04 | 0.33±0.10 | 0.04±0.01 | 0.05±0.02 |
| 15 | RememberColor3-v0 | 0.65±0.04 | 0.01±0.01 | 0.27±0.03 | 0.29±0.01 | 0.32±0.01 |
| 16 | RememberColor5-v0 | 0.13±0.03 | 0.07±0.05 | 0.12±0.01 | 0.15±0.02 | 0.10±0.02 |
| 17 | RememberColor9-v0 | 0.09±0.02 | 0.01±0.01 | 0.12±0.02 | 0.15±0.01 | 0.17±0.01 |
| 18 | RememberShape3-v0 | 0.21±0.04 | 0.05±0.04 | 0.31±0.04 | 0.20±0.10 | 0.32±0.05 |
| 19 | RememberShape5-v0 | 0.17±0.04 | 0.04±0.04 | 0.18±0.01 | 0.15±0.00 | 0.21±0.04 |
| 20 | RememberShape9-v0 | 0.05±0.00 | 0.05±0.02 | 0.10±0.02 | 0.14±0.01 | 0.11±0.02 |
| 21 | RememberShapeAndColor3x2-v0 | 0.14±0.02 | 0.04±0.02 | 0.13±0.02 | 0.11±0.05 | 0.14±0.02 |
| 22 | RememberShapeAndColor3x3-v0 | 0.08±0.03 | 0.06±0.06 | 0.09±0.02 | 0.09±0.02 | 0.16±0.01 |
| 23 | RememberShapeAndColor5x3-v0 | 0.07±0.02 | 0.01±0.01 | 0.09±0.01 | 0.09±0.02 | 0.11±0.03 |
| 24 | BunchOfColors3-v0 | 0.00±0.00 | 0.00±0.00 | 0.00±0.00 | 0.00±0.00 | 0.00±0.00 |
| 25 | BunchOfColors5-v0 | 0.00±0.00 | 0.00±0.00 | 0.00±0.00 | 0.00±0.00 | 0.00±0.00 |
| 26 | BunchOfColors7-v0 | 0.00±0.00 | 0.00±0.00 | 0.00±0.00 | 0.00±0.00 | 0.00±0.00 |
| 27 | SeqOfColors3-v0 | 0.00±0.00 | 0.00±0.00 | 0.00±0.00 | 0.00±0.00 | 0.00±0.00 |
| 28 | SeqOfColors5-v0 | 0.00±0.00 | 0.00±0.00 | 0.00±0.00 | 0.00±0.00 | 0.00±0.00 |
| 29 | SeqOfColors7-v0 | 0.00±0.00 | 0.00±0.00 | 0.00±0.00 | 0.00±0.00 | 0.00±0.00 |
| 30 | ChainOfColors3-v0 | 0.00±0.00 | 0.00±0.00 | 0.00±0.00 | 0.00±0.00 | 0.00±0.00 |
| 31 | ChainOfColors5-v0 | 0.00±0.00 | 0.00±0.00 | 0.00±0.00 | 0.00±0.00 | 0.00±0.00 |
| 32 | ChainOfColors7-v0 | 0.00±0.00 | 0.00±0.00 | 0.00±0.00 | 0.00±0.00 | 0.00±0.00 |

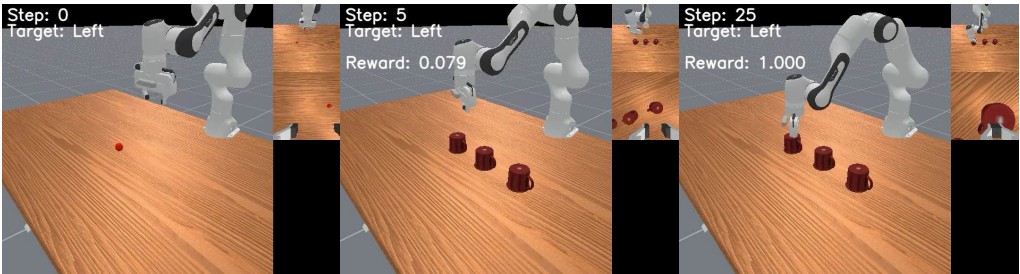

Figure 14: `ShellGameTouch-v0`: The robot observes a ball in front of it. next, this ball is covered by a mug and then the robot has to touch the mug with the ball underneath.

## K.1 SHELLGAME-V0

The `ShellGame-v0` task (Figure 14) is inspired by a simplified version of the classic shell game, which tests a person's ability to remember object locations when they become occluded. This task evaluates an agent's capacity for object permanence and spatial memory, crucial skills for real-world robotic manipulation where objects frequently become temporarily hidden from view.

**Environment Description**    The environment consists of three identical mugs placed on a table and a red ball. The task proceeds in three phases:

1. **Observation Phase** (steps 0-4): The ball is placed at one of three positions, and the agent can observe its location.
2. **Occlusion Phase** (step 5): The ball and positions are covered by three identical mugs.
3. **Action Phase** (steps 6+): The agent must interact with the mug covering the ball's location. The type of target interaction depends on the selected mode: `Touch`, `Push` and `Pick`.

**Task Modes**    The task includes three variants of increasing difficulty:

- `Touch`: The agent only needs to touch the correct mug
- `Push`: The agent must push the correct mug to a designated area
- `Pick`: The agent must pick and lift the correct mug above a specified height

**Success Criteria**    Success is determined by:

- `Touch`: Contact between the gripper and the correct mug
- `Push`: Moving forward the correct mug to the target zone
- `Pick`: Elevating the correct mug above 0.1m

**Reward Structure**    The environment provides both sparse and dense reward variants:

- **Sparse**: Binary reward (1.0 for success, 0.0 otherwise)
- **Dense**: Continuous reward based on:
    - Distance between gripper and target mug
    - Robot's motion smoothness (static reward based on joint velocities)
    - Task completion status (additional reward when the task is solved)

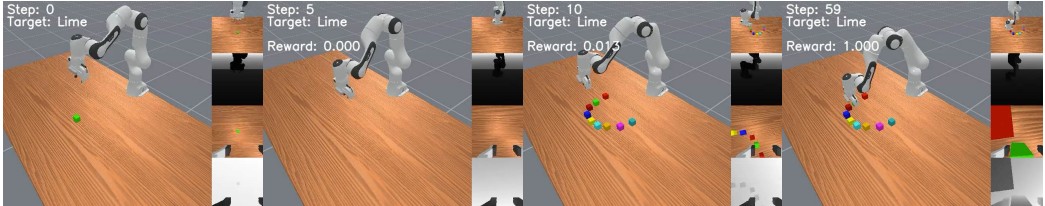

Figure 15: `RememberColor9-v0`: The robot observes a colored cube in front of it, then this cube disappears and an empty table is shown. Then 9 cubes appear on the table, and the agent must touch a cube of the same color as the one it observed at the beginning of the episode.

### K.2 REMEMBERCOLOR-V0

The `RememberColor-v0` task (Figure 15) tests an agent's ability to remember and identify objects based on their visual properties. This capability is essential for real-world robotics applications where agents must recall and match object characteristics across time intervals.

**Environment Description** The environment presents a sequence of colored cubes on a table. The task proceeds in three phases:

1. **Observation Phase** (steps 0-4): A cube of a specific color is displayed, and the agent must memorize its color.
2. **Delay Phase** (steps 5-9): The cube disappears, leaving an empty table.
3. **Selection Phase** (steps 10+): Multiple cubes of different colors appear (3, 5, or 9 depending on difficulty), and the agent must identify and interact with the cube matching the original color.

**Task Modes** The task includes three complexity levels:

- `3` (easy): Choose from 3 different colors (red, lime, blue)
- `5` (Medium): Choose from 5 different colors (red, lime, blue, yellow, magenta)
- `9` (Hard): Choose from 9 different colors (red, lime, blue, yellow, magenta, cyan, maroon, olive, teal)

**Success Criteria** Success is determined by:

- Correctly identifying and touching the cube that matches the color shown in the observation phase
- Maintaining contact with the correct cube for at least 0.1 seconds

**Reward Structure** The environment provides both sparse and dense reward variants:

- **Sparse**: Binary reward (1.0 for success, 0.0 otherwise)
- **Dense**: Continuous reward based on:
    - Distance between gripper and target cube
    - Robot's motion smoothness (static reward based on joint velocities)
    - Additional reward for robot being static while touching the correct cube
    - Task completion status (additional reward when the task is solved)

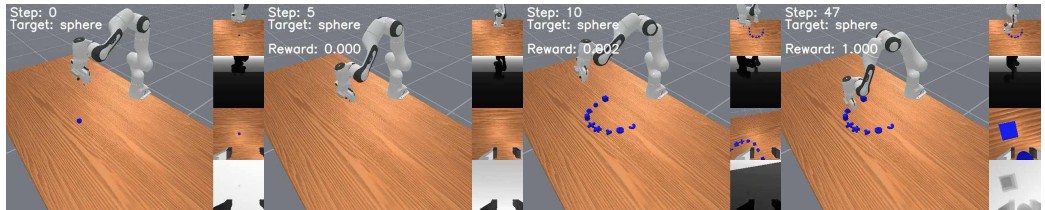

Figure 16: `RememberShape9-v0`: The robot observes an object with specific shape in front of it, then the object disappears and an empty table appears. Then 9 objects of different shapes appear on the table, and the agent must touch an object of the same shape as the one it observed at the beginning of the episode.

### K.3 REMEMBERSHAPE-V0

The `RememberShape-v0` task (Figure 16) evaluates an agent's ability to remember and identify objects based on their geometric properties. This capability is crucial for robotic applications where shape recognition and recall are essential for successful manipulation.

**Environment Description**   The environment presents a sequence of geometric shapes on a table. The task proceeds in three phases:

1. **Observation Phase** (steps 0-4): A shape (cube, sphere, cylinder, etc.) is displayed, and the agent must memorize its geometry.
2. **Delay Phase** (steps 5-9): The shape disappears, leaving an empty table.
3. **Selection Phase** (steps 10+): Multiple shapes appear (3, 5, or 9 depending on difficulty), and the agent must identify and interact with the shape matching the original geometry.

**Task Modes**   The task includes three complexity levels:

- `3` (Easy): Choose from 3 different shapes (cube, sphere, cylinder)
- `5` (Medium): Choose from 5 different shapes (cube, sphere, cylinder cross, torus)
- `9` (Hard): Choose from 9 different shapes (cube, sphere, cylinder cross, torus, star, pyramid, t-shape, crescent)

**Success Criteria**   Success is determined by:

- Correctly identifying and touching the object with the same shape shown in the observation phase
- Maintaining contact with the correct shape for at least 0.1 seconds

**Reward Structure**   The environment provides both sparse and dense reward variants:

- **Sparse**: Binary reward (1.0 for success, 0.0 otherwise)
- **Dense**: Continuous reward based on:
    - Distance between gripper and target object
    - Robot's motion smoothness (static reward based on joint velocities)
    - Additional reward for maintaining static position when touching correct object
    - Task completion status (additional reward when the task is solved)

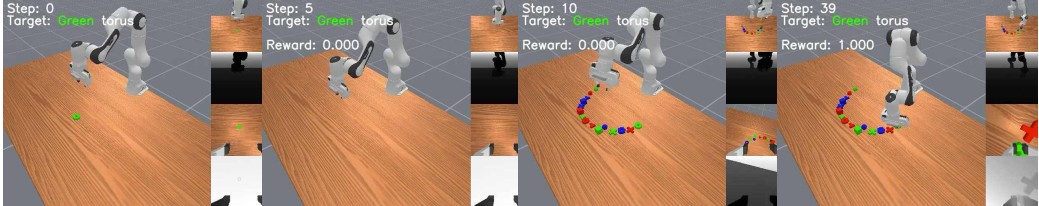

Figure 17: `RememberShapeAndColor5x3-v0`: An object of a certain shape and color appears in front of the agent. Then the object disappears and the agent sees an empty table. Then objects of 5 different shapes and 3 different colors appear on the table and the agent has to touch what it observed at the beginning of the episode.

## K.4 REMEMBERSHAPEANDCOLOR-V0

The `RememberShapeAndColor-v0` task (Figure 17) evaluates an agent's ability to remember and identify objects based on multiple visual properties simultaneously. This task combines shape and color recognition, testing the agent's capacity to maintain and match multiple object features across time intervals.

**Environment Description** The environment presents a sequence of colored geometric shapes on a table. The task proceeds in three phases:

1. **Observation Phase** (steps 0-4): An object with specific shape and color is displayed, and the agent must memorize both properties.
2. **Delay Phase** (steps 5-9): The object disappears, leaving an empty table.
3. **Selection Phase** (steps 10+): Multiple objects with different combinations of shapes and colors appear, and the agent must identify and interact with the object matching both the original shape and color.

**Task Modes** The task includes three complexity levels based on the number of shape-color combinations:

- `3x2` (Easy): Choose from 6 objects (3 shapes × 2 colors); shapes: cube, sphere, t-shape; colors: red, green
- `3x3` (Medium): Choose from 9 objects (3 shapes × 3 colors); shapes: cube, sphere, t-shape; colors: red, green, blue
- `5x3` (Hard): Choose from 15 objects (5 shapes × 3 colors); shapes: cube, sphere, t-shape, cross, torus; colors: red, green, blue

**Success Criteria** Success is determined by:

- Correctly identifying and touching the object that matches both the shape and color shown in the observation phase
- Maintaining contact with the correct object for at least 0.1 seconds

**Reward Structure** The environment provides both sparse and dense reward variants:

- **Sparse**: Binary reward (1.0 for success, 0.0 otherwise)
- **Dense**: Continuous reward based on:
  - Distance between gripper and target object
  - Robot's motion smoothness (static reward based on joint velocities)
  - Additional reward for maintaining static position while touching correct object
  - Task completion status (additional reward when the task is solved)

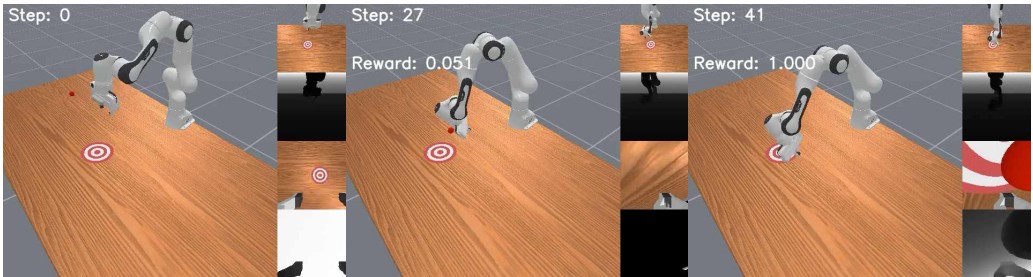

Figure 18: `InterceptMedium-v0`: A ball rolls on the table in front of the agent with a random initial velocity, and the agent's task is to intercept this ball and direct it at the target zone.

### K.5 INTERCEPT-V0

The `Intercept-v0` task (Figure 19) evaluates an agent's ability to predict and intercept a moving object based on its initial trajectory. This task tests the agent's capacity for motion prediction and spatial-temporal reasoning, which are essential skills for dynamic manipulation tasks in robotics.

**Environment Description**  The environment consists of a red ball moving across a table and a target zone. The task requires the agent to:

1. Observe the ball's initial position and velocity
2. Predict the ball's trajectory
3. Guide the ball to reach a designated target zone

**Task Modes**  The task includes three variants with increasing ball velocities:

- `Slow`: Ball velocity range of 0.25-0.5 m/s
- `Medium`: Ball velocity range of 0.5-0.75 m/s
- `Fast`: Ball velocity range of 0.75-1.0 m/s

**Success Criteria**  Success is determined by:

- Guiding the ball to enter the target zone
- The ball must come to rest within the target area (radius 0.1m)

**Reward Structure**  The environment provides both sparse and dense reward variants:

- **Sparse**: Binary reward (1.0 for success, 0.0 otherwise)
- **Dense**: Continuous reward based on:
    - Distance between gripper and ball
    - Distance between ball and target zone
    - Robot's motion smoothness (static reward based on joint velocities)
    - Task completion status (additional reward when the task is solved)

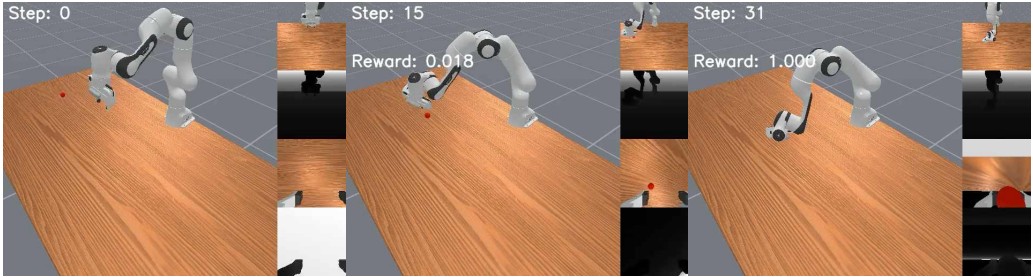

Figure 19: `InterceptGrabMedium-v0`: A ball rolls on the table in front of the agent with a random initial velocity, and the agent's task is to intercept this ball with a gripper and lift it up.

### K.6 INTERCEPTGRAB-V0

The `InterceptGrab-v0` task (Figure 19) extends the `Intercept-v0` task by requiring the agent to not only predict the trajectory of a moving object but also grasp it while in motion. This task evaluates the agent's ability to combine motion prediction with precise manipulation timing, simulating real-world scenarios where robots must catch or intercept moving objects.

**Environment Description** The environment consists of a red ball moving across a table. The task requires the agent to:

1. Observe the ball's initial position and velocity
2. Predict the ball's trajectory
3. Position the gripper to intercept the ball's path
4. Time the grasping action correctly to catch the ball
5. Maintain a stable grasp while bringing the ball to rest

**Task Modes** The task includes three variants with increasing ball velocities:

- `Slow`: Ball velocity range of 0.25-0.5 m/s
- `Medium`: Ball velocity range of 0.5-0.75 m/s
- `Fast`: Ball velocity range of 0.75-1.0 m/s

**Success Criteria** Success is determined by:

- Successfully grasping the moving ball
- Maintaining a stable grasp until the ball comes to rest
- The robot must be static with the ball firmly grasped

**Reward Structure** The environment provides both sparse and dense reward variants:

- **Sparse**: Binary reward (1.0 for success, 0.0 otherwise)
- **Dense**: Continuous reward based on:
    - Distance between gripper and ball
    - Grasping reward
    - Robot's motion smoothness (static reward based on joint velocities)
    - Task completion status (additional reward when the task is solved)

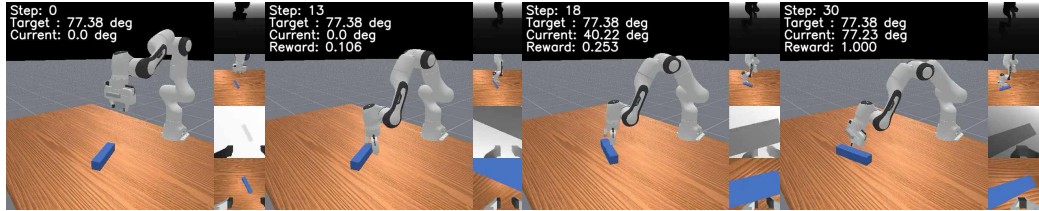

Figure 20: `RotateLenientPos-v0`: A randomly oriented peg is placed in front of the agent. The agent's task is to rotate this peg by a certain angle (the center of the peg can be shifted).

### K.7 ROTATELENIENT-V0

The `RotateLenient-v0` task (Figure 20) evaluates an agent's ability to remember and execute specific rotational movements. This task tests the agent's capacity to maintain and reproduce angular information, which is crucial for manipulation tasks requiring precise orientation control. This task tests the agent's ability to hold information in memory about how far peg has already rotated at the current step relative to its initial position.

**Environment Description**   The environment consists of a blue-colored peg on a table that must be rotated by a specified angle. The task proceeds in one phase, but the static prompt information about the target angle is available to the agent at each timestep:

1. **Action Phase**: The agent must rotate the peg to match the target angle

**Task Modes**   The task includes two variants with different rotation requirements:

- `Pos`: Rotate by a positive angle between 0 and $\pi/2$
- `PosNeg`: Rotate by either positive or negative angle between $-\pi/4$ and $\pi/4$

**Success Criteria**   Success is determined by:

- Rotating the peg to within the angle threshold (±0.1 radians) of the target angle
- Maintaining the final orientation in a stable position
- The robot must be static with the peg at the correct orientation

**Reward Structure**   The environment provides both sparse and dense reward variants:

- **Sparse**: Binary reward (1.0 for success, 0.0 otherwise)
- **Dense**: Continuous reward based on:
  - Distance between gripper and peg
  - Angular distance to target rotation
  - Stability of the peg's orientation
  - Robot's motion smoothness (static reward based on joint velocities)
  - Task completion status (additional reward when the task is solved)

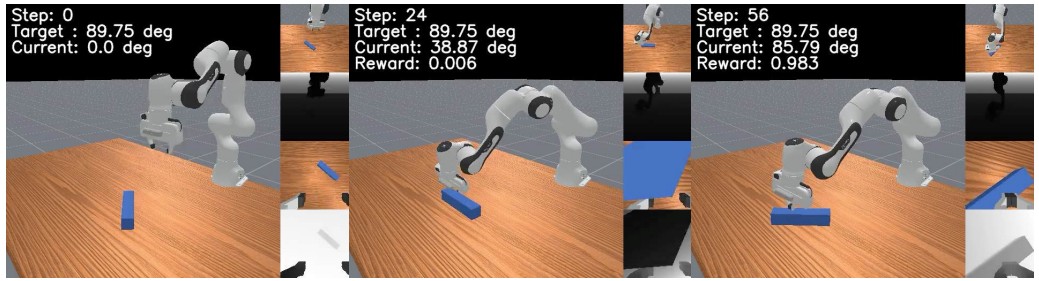

Figure 21: `RotateStrictPos-v0`: A randomly oriented peg is placed in front of the agent. The agent's task is to rotate this peg by a certain angle (it is not allowed to move the center of the peg)

### K.8   ROTATESTRICT-V0

The `RotateStrict-v0` task (Figure 21) extends the `RotateLenient-v0` task with more stringent requirements for precise rotational control.

**Environment Description**   The environment consists of a blue-colored peg on a table that must be rotated by a specified angle while maintaining its position. The task proceeds in one phase, but the static prompt information about the target angle is available to the agent at each timestep:

1. **Action Phase**: The agent must rotate the peg to match the target angle while keeping it centered

**Task Modes**   The task includes two variants with different rotation requirements:

- `Pos`: Rotate by a positive angle between 0 and $\pi/2$
- `PosNeg`: Rotate by either positive or negative angle between $-\pi/4$ and $\pi/4$

**Success Criteria**   Success is determined by:

- Rotating the peg to within the angle threshold (±0.1 radians) of the target angle
- Maintaining the peg's position within 5cm of its initial XY coordinates
- The robot must be static with the peg at the correct orientation
- No significant deviation in other rotation axes

**Reward Structure**   The environment provides both sparse and dense reward variants:

- **Sparse**: Binary reward (1.0 for success, 0.0 otherwise)
- **Dense**: Continuous reward based on:
    - Distance between gripper and peg
    - Angular distance to target rotation
    - Position deviation from initial location
    - Stability of the peg's orientation
    - Robot's motion smoothness (static reward based on joint velocities)
    - Task completion status (additional reward when the task is solved)

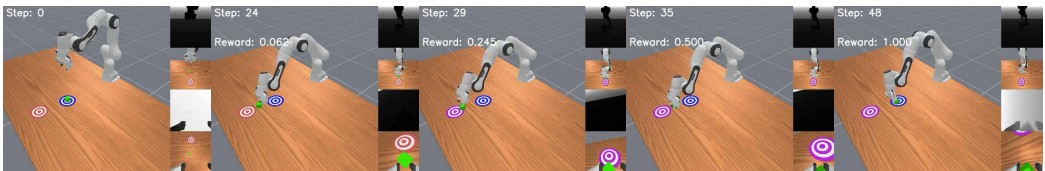

Figure 22: `TakeItBack-v0`: The agent observes a green cube in front of him. The agent's task is to move the green cube to the red target, and as soon as it lights up violet, return the cube to its original position (the agent does not observes the original position of the cube).

### K.9   TAKEITBACK-V0

The `TakeItBack-v0` task (Figure 22) assesses the agent's ability to perform sequential tasks and memorize the starting position. This task tests the agent's capacity for sequential memory and spatial reasoning, requiring it to maintain information about past locations and achievements while executing a multi-step plan.

**Environment Description**   The environment consists of a green cube and two target regions (initial and goal) on a table. The task proceeds in two phases:

1. **First Phase**: The agent must move the cube from its initial position to a goal region
2. **Second Phase**: After reaching the goal, goal region change it's color from red to magenta, and the agent must return the cube to its original position (marked by the initial region and invisible for the agent)

**Success Criteria**   Success is determined by:

- First reaching the goal region with the cube
- Then returning the cube to the initial region
- Both goals must be achieved in sequence

**Reward Structure**   The environment provides both sparse and dense reward variants:

- **Sparse**: Binary reward (1.0 for success, 0.0 otherwise)
- **Dense**: Continuous reward based on:
    - Distance between gripper and cube
    - Distance to current target region
    - Progress through the task sequence
    - Stability of cube manipulation
    - Robot's motion smoothness (static reward based on joint velocities)
    - Task completion status (additional reward when the task is solved)

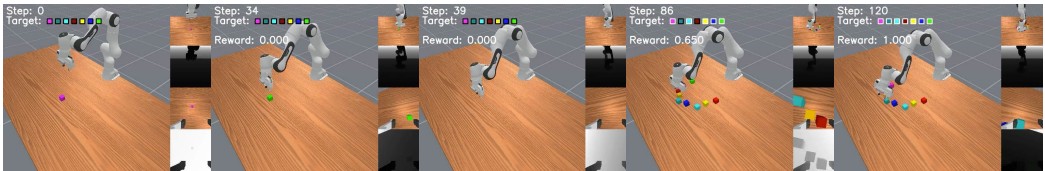

Figure 23: `SeqOfColors7-v0`: In front of the agent, 7 cubes of different colors appear sequentially. After the last cube is shown, the agent observes an empty table. Then 9 cubes of different colors appear on the table and the agent has to touch the cubes that were shown at the beginning of the episode in any order.

## K.10 SEQOFCOLORS-V0

The `SeqOfColors-v0` task (Figure 23) evaluates an agent's ability to remember and reproduce an unordered sequence of colors. This task tests memory capacity capabilities essential for robotic tasks that require following specific patterns or sequences.

**Environment Description**    The environment presents a sequence of colored cubes that must be reproduced in any order. The task proceeds in two phases:

1. **Observation Phase** (steps 0-$(5N-1)$): A sequence of N colored cubes is shown one at a time, with each cube visible for 5 steps.
2. **Delay Phase** (steps $(5N)$-$(5N+4)$): All cubes disappear
3. **Selection Phase** (steps $(5N+5)$+): A larger set of cubes appears, and the agent must identify and touch all previously shown cubes in any order

**Task Modes**    The task includes three complexity levels:

- 3 (Easy): Remember 3 colors demonstrated sequentially
- 5 (Medium): Remember 5 colors demonstrated sequentially
- 7 (Hard): Remember 7 colors demonstrated sequentially

**Success Criteria**    Success is determined by:

- Correctly identifying and touching all cubes from the observation phase
- Order of selection doesn't matter
- Each cube must be touched for at least 0.1 seconds
- The demonstrated set must be touched without any mistakes

**Reward Structure**    The environment provides both sparse and dense reward variants:

- **Sparse**: Binary reward (1.0 for success, 0.0 otherwise)
- **Dense**: Continuous reward based on:
    - Distance between gripper and next target cube
    - Number of correctly identified cubes
    - Static reward for stable contact
    - Robot's motion smoothness (static reward based on joint velocities)
    - Task completion status (additional reward when the task is solved)

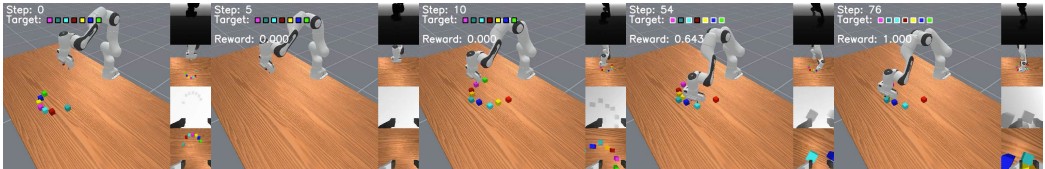

Figure 24: `BunchOfColors7-v0`: 7 cubes of different colors appear simultaneously in front of the agent. After the agent observes an empty table. Then, 9 cubes of different colors appear on the table and the agent has to touch the cubes that were shown at the beginning of the episode in any order.

### K.11  BUNCHOFCOLORS-V0

The `BunchOfColors-v0` task (Figure 24) tests an agent's memory capacity by requiring it to remember multiple objects simultaneously. This capability is crucial for tasks requiring parallel processing of multiple items.

**Environment Description**  The environment presents multiple colored cubes simultaneously. The task proceeds in three phases:

1. **Observation Phase** (steps 0-4): Multiple colored cubes are displayed simultaneously
2. **Delay Phase** (steps 5-9): All cubes disappear
3. **Selection Phase** (steps 10+): A larger set of cubes appears, and the agent must identify and touch all previously shown cubes in any order

**Task Modes**  The task includes three complexity levels:

- 3 (Easy): Remember 3 colors demonstrated simultaneously
- 5 (Medium): Remember 5 colors demonstrated simultaneously
- 7 (Hard): Remember 7 colors demonstrated simultaneously

**Success Criteria**  Success is determined by:

- Correctly identifying and touching all cubes from the observation phase
- Order of selection doesn't matter
- Each cube must be touched for at least 0.1 seconds
- The demonstrated set must be touched without any mistakes

**Reward Structure**  The environment provides both sparse and dense reward variants:

- **Sparse**: Binary reward (1.0 for success, 0.0 otherwise)
- **Dense**: Continuous reward based on:
    - Distance between gripper and next target cube
    - Static reward for stable contact
    - Number of correctly touched cubes
    - Robot's motion smoothness (static reward based on joint velocities)
    - Task completion status (additional reward when the task is solved)

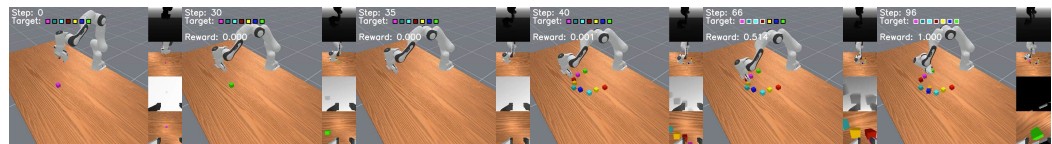

Figure 25: `ChainOfColors7-v0`: In front of the agent, 7 cubes of different colors appear sequentially. After the last cube is shown, the agent sees an empty table. Then 9 cubes of different colors appear on the table and the agent must unmistakably touch the cubes that were shown at the beginning of the episode, in the same strict order.

### K.12   CHAINOFCOLORS-V0

The `ChainOfColors-v0` task (Figure 25) evaluates the agent's ability to store and retrieve ordered information. This task simulates scenarios where the agent must track changing relationships between objects over time.

**Environment Description**   The environment presents am ordered sequence (chain) of colored cubes that must be followed. The task proceeds in multiple phases:

1. **Observation Phase** (steps 0-$(5N-1)$): A sequence of N colored cubes is shown one at a time, with each cube visible for 5 steps.
2. **Delay Phase** (steps $(5N)$-$(5N+4)$): All cubes disappear
3. **Selection Phase** (steps $(5N+5)$+): A larger set of cubes appears, and the agent must identify and touch all previously shown cubes in the exact order as demonstrated

**Task Modes**   The task includes three complexity levels:

- 3 (Easy): Remember 3 colors demonstrated sequentially
- 5 (Medium): Remember 5 colors demonstrated sequentially
- 7 (Hard): Remember 7 colors demonstrated sequentially

**Success Criteria**   Success is determined by:

- Correctly identifying and touching all cubes from the observation phase in the exact order
- Each cube must be touched for at least 0.1 seconds
- The demonstrated set must be touched without any mistakes

**Reward Structure**   The environment provides both sparse and dense reward variants:

- **Sparse**: Binary reward (1.0 for success, 0.0 otherwise)
- **Dense**: Continuous reward based on:
  - Distance between gripper and next target cube
  - Static reward for stable contact
  - Number of correctly touched cubes
  - Robot's motion smoothness (static reward based on joint velocities)
  - Task completion status (additional reward when the task is solved)

Table 9: Classification of environments from the MIKASA-Base benchmark according to the suggested memory-intensive tasks classification from the Section 4.2.

| Environment | Memory Task | Brief description of the task | Observation Space | Action Space |
|---|---|---|---|---|
| Memory Cards | Capacity | Memorize the positions of revealed cards and correctly match pairs while minimizing incorrect guesses. | vector | discrete |
| Numpad | Sequential | Memorize the sequence of movements and navigate the rolling ball on a 3×3 grid by following the correct order while avoiding mistakes. | image, vector | discrete, continuous |
| BSuite Memory Length | Object | Memorize the initial context signal and recall it after a given number of steps to take the correct action. | vector | discrete |
| Minigrid-Memory | Object | Memorize the object in the starting room and use this information to select the correct path at the junction. | image | discrete |
| Ballet | Sequential, Object | Memorize the sequence of movements performed by each uniquely colored and shaped dancer, then identify and approach the dancer who executed the given pattern. | image | discrete |
| Passive Visual Match | Object | Memorize the target color displayed on the wall during the initial phase. After a brief distractor phase, identify and select the target color among the distractors by stepping on the corresponding ground pad. | image | discrete |
| Passive-T-Maze | Object | Memorize the goal's location upon initial observation, navigate through the maze with limited sensory input, and select the correct path at the junction. | vector | discrete |
| ViZDoom-two-colors | Object | Memorize the color of the briefly appearing pillar (green or red) and collect items of the same color to survive in the acid-filled room. | image | discrete |
| Memory Maze | Spatial | Memorize the locations of objects and the maze structure using visual clues, then navigate efficiently to find objects of a specific color and score points. | image | discrete |
| MemoryGym Mortar Mayhem | Capacity, Sequential | Memorize a sequence of movement commands and execute them in the correct order. | image | discrete |
| MemoryGym Mystery Path | Capacity, Spatial | Memorize the invisible path and navigate it without stepping off. | image | discrete |
| POPGym Repeat First | Object | Memorize the initial value presented at the first step and recall it correctly after receiving a sequence of random values. | vector | discrete |
| POPGym Repeat Previous | Sequential, Object | Memorize the value observed at each step and recall the value from $k$ steps earlier when required. | vector | discrete |
| POPGym Autoencode | Sequential | Memorize the sequence of cards presented at the beginning and reproduce them in the same order when required. | vector | discrete |
| POPGym Count Recall | Object, Capacity | Memorize unique values encountered and count how many times a specific value has appeared. | vector | discrete |
| POPGym vectorless Cartpole | Sequential | Memorize velocity data over time and integrate it to infer the position of the pole for balance control. | vector | continuous |
| POPGym vectorless Pendulum | Sequential | Memorize angular velocity over time and integrate it to infer the pendulum's position for successful swing-up control. | vector | continuous |
| POPGym Multiarmed Bandit | Object, Capacity | Memorize the reward probabilities of different slot machines by exploring them and identify the one with the highest expected reward. | vector | discrete |
| POPGym Concentration | Capacity | Memorize the positions of revealed cards and match them with previously seen cards to find all matching pairs. | vector | discrete |
| POPGym Battleship | Spatial | Memorize the coordinates of previous shots and their HIT or MISS feedback to build an internal representation of the board, avoid repeat shots, and strategically target ships for maximum rewards. | vector | discrete |
| POPGym Mine Sweeper | Spatial | Memorize revealed grid information and use numerical clues to infer safe tiles while avoiding mines. | vector | discrete |
| POPGym Labyrinth Explore | Spatial | Memorize previously visited cells and navigate the maze efficiently to discover new, unexplored areas and maximize rewards. | vector | discrete |
| POPGym Labyrinth Escape | Spatial | Memorize the maze layout while exploring and navigate efficiently to find the exit and receive a reward. | vector | discrete |
| POPGym Higher Lower | Object, Sequential | Memorize previously revealed card ranks and predict whether the next card will be higher or lower, updating the reference card after each prediction to maximize rewards. | vector | discrete |

# L   MIKASA-BASE BENCHMARK TASKS DESCRIPTION

This section provides a detailed description of all environments included in the MIKASA-Base benchmark Section 5. Understanding the characteristics and challenges of these environments is crucial for evaluating RL algorithms. Each environment presents unique tasks, offering diverse scenarios to test the memory abilities of RL agents.

## L.1   MEMORY CARDS

The Memory Cards environment (Esslinger et al., 2022) is a memory game environment with 5 randomly shuffled pairs of hidden cards. At each step, the agent sees one revealed card and must find its matching pair. A correct guess removes both cards; otherwise, the card is hidden again, and a new one is revealed. The game continues until all pairs are removed.

## L.2   NUMPAD

The Numpad environment (Humplik et al., 2019) consists of an $N \times N$ grid of tiles. The agent controls a ball that rolls between tiles. At the beginning of an episode, a random sequence of $n$ neighboring tiles (excluding diagonals) is selected, and the agent must follow this sequence in the correct order. The environment is structured so that pressing the correct tile lights it up, while pressing an incorrect tile resets progress. A reward of +1 is given for the first press of each correct tile after a reset. The episode ends after a fixed number of steps. To succeed, the agent must memorize the sequence and navigate it correctly without mistakes. The ability to "jump" over tiles is not available.

### L.3  BSUITE MEMORYLENGTH

The MemoryLength environment (Osband et al., 2020) represents a sequence of observations, where at each step, the observation takes a value of either +1 or -1. The environment is structured so that a reward is given only at the final step if the agent correctly predicts the $i$-th value from the initial observation vector $obs$. The index of this $i$-th value is specified at the last step observation vector in $obs[1]$. To succeed, the agent must remember the sequence of observations and use this information to make an accurate prediction at the final step.

### L.4  MINIGRID-MEMORY

Minigrid-Memory (Chevalier-Boisvert et al., 2023) is a two-dimensional grid-based environment that features a T-shaped maze with a small room at the beginning of the corridor, containing an object. The agent starts at a random position within the corridor. Its task is to reach the room, observe and memorize the object, then proceed to the junction at the maze's end and turn towards the direction where an identical object is located. The reward function is defined as $R_t = 1 - 0.9 \times \frac{t}{T}$ for a successful attempt; otherwise, the agent receives zero reward. The episode terminates when the agent makes a choice at the junction or exceeds a time limit of steps.

### L.5  BALLET

In the Ballet environment (Lampinen et al., 2021) tasks take place in an $11 \times 11$ tiled room, consisting of a $9 \times 9$ central area surrounded by a one-tile-wide wall. Each tile is upsampled to 9 pixels, resulting in a $99 \times 99$ pixel input image. The agent is initially placed at the center of the room, while dancers are randomly positioned in one of 8 possible locations around it. Each dancer has a distinct shape and color, selected from 15 possible shapes and 19 colors, ensuring uniqueness. These visual features serve only for identification and do not influence behavior. The agent itself is always represented as a white square. The agent receives egocentric visual observations, meaning its view is centered on its own position, which has been shown to enhance generalization.

### L.6  PASSIVE T-MAZE

The Passive T-Maze environment (Ni et al., 2023) consists of a corridor leading to a junction that connects two possible goal states. The agent starts at a designated position and can move in four directions: left, right, up, or down. At the beginning of each episode, one of the two goal states is randomly assigned as the correct destination. The agent observes this goal location before starting its movement. The agent stays in place if it attempts to move into a wall. To succeed, the agent must navigate to the correct goal based on its initial observation. The optimal strategy involves moving through the corridor towards the junction and then selecting the correct path.

### L.7  VIZDOOM-TWO-COLORS

The ViZDoom-Two-Colors (Sorokin et al., 2022) is an environment where an agent is placed in a room with constantly depleting health. The room contains red and green objects, one of which restores health (+1 reward), while the other reduces it (-1 reward). The beneficial color is randomly assigned at the beginning of each episode and indicated by a column. The environment is structured so that the agent must memorize the column's color to collect the correct items. Initially, the column remains visible, but in a harder variant, it disappears after 45 steps, increasing the memory requirement. To succeed, the agent must maximize survival by collecting beneficial objects while avoiding harmful ones.

### L.8  MEMORY MAZE

The Memory Maze environment Pasukonis et al. (2022) is a procedurally generated 3D maze. Each episode, the agent spawns in a new maze with multiple colored objects placed in fixed locations. The agent receives a first-person view and a prompt indicating the color of the target object. It must navigate the maze, memorize object positions, and return to them efficiently. The agent receives a reward of 1 for reaching the correct object and no reward for incorrect objects.

### L.9 MEMORYGYM MORTAR MAYHEM

Mortar Mayhem (Pleines et al., 2023) is a grid-based environment where the agent must memorize and execute a sequence of commands in the correct order. The environment consists of a finite grid, where the agent initially observes a series of movement instructions and then attempts to reproduce them accurately. Commands include movements to adjacent tiles or remaining in place. The challenge lies in the agent's ability to recall and execute a growing sequence of instructions, with failure resulting in episode termination. A reward of +0.1 is given for each correctly executed command

### L.10 MEMORYGYM MYSTERY PATH

Mystery Path (Pleines et al., 2023) presents an invisible pathway that the agent must traverse without deviating. If the agent steps off the path, it is returned to the starting position, forcing it to remember the correct trajectory. The path is procedurally generated, meaning each episode introduces a new configuration. Success in this environment requires the agent to accurately recall previous steps and missteps to avoid repeating errors. The agent is rewarded +0.1 for progressing onto a previously unvisited tile

### L.11 POPGYM ENVIRONMENTS

The following environments are included from the POPGym benchmark (Morad et al., 2023), which is designed to evaluate RL agents in partially observable settings. POPGym provides a diverse collection of lightweight vectorized environments with varying difficulty levels.

#### L.11.1 POPGYM AUTOENCODE

The environment consists of a deck of cards that is shuffled and sequentially shown to the agent during the watch phase. While observing the cards, a watch indicator is active, but it disappears when the last card is revealed. Afterward, the agent must reproduce the sequence of cards in the correct order. The environment is structured to evaluate the agent's ability to encode a sequence of observations into an internal representation and later reconstruct the sequence one observation at a time.

#### L.11.2 POPGYM CONCENTRATION

The environment represents a classic memory game where a shuffled deck of cards is placed face-down. The agent sequentially flips two cards and earns a reward if the revealed cards form a matching pair. The environment is designed in such a way that the agent must remember previously revealed cards to maximize its success rate.

#### L.11.3 POPGYM REPEAT FIRST

The environment presents the agent with an initial value from a set of four possible values, along with an indicator signaling that this is the first value. In subsequent steps, the agent continues to receive random values from the same set but without the initial indicator. The structure requires the agent to retain the first received value in memory and recall it accurately to receive a reward.

#### L.11.4 POPGYM REPEAT PREVIOUS

The environment consists of a sequence of observations, where each observation can take one of four possible values at each timestep. The agent is tasked with recalling and outputting the value that appeared a specified number of steps in the past.

#### L.11.5 POPGYM STATELESS CARTPOLE

This is a modified version of the traditional Cartpole environment (Barto et al., 1983) where angular and linear position information is removed from observations. Instead, the agent only receives velocity-based data and must infer positional states by integrating this information over time to successfully balance the pole.

### L.11.6   POPGYM STATELESS PENDULUM

In this variation of the swing-up pendulum environment (Doya, 1995), angular position data is omitted from the agent's observations. The agent must infer the pendulum's position by processing velocity information and use this estimate to determine appropriate control actions.

### L.11.7   POPGYM NOISY STATELESS CARTPOLE

This environment builds upon Stateless Cartpole by introducing Gaussian noise into the observations. The agent must still infer positional states from velocity information while filtering out the added noise to maintain control of the pole.

### L.11.8   POPGYM NOISY STATELESS PENDULUM

This variation extends the Stateless Pendulum environment by incorporating Gaussian noise into the observations. The agent must manage this uncertainty while using velocity data to estimate the pendulum's position and swing it up effectively.

### L.11.9   POPGYM MULTIARMED BANDIT

The Multiarmed Bandit environment is an episodic formulation of the multiarmed bandit problem (Slivkins, 2024), where a set of bandits is randomly initialized at the start of each episode. Unlike conventional multiarmed bandit tasks, where reward probabilities remain fixed across episodes, this structure resets them every time. The agent must dynamically adjust its exploration and exploitation strategies to maximize long-term rewards.

### L.11.10   POPGYM HIGHER LOWER

Inspired by the higher-lower card game, this environment presents the agent with a sequence of cards. At each step, the agent must predict whether the next card will have a higher or lower rank than the current one. Upon making a guess, the next card is revealed and becomes the new reference. The agent can enhance its performance by employing card counting strategies to estimate the probability of future values.

### L.11.11   POPGYM COUNT RECALL

At each timestep, the agent is presented with two values: a next value and a query value. The agent must determine and output how many times the query value has appeared so far. To succeed, the agent must maintain an accurate count of past occurrences and retrieve the correct number upon request.

### L.11.12   POPGYM BATTLESHIP

A partially observable variation of the game Battleship, where the agent does not have access to the full board. Instead, it receives feedback on its previous shot, indicating whether it was a HIT or MISS, along with the shot's location. The agent earns rewards for hitting ships, receives no reward for missing, and incurs a penalty for targeting the same location more than once. The environment challenges the agent to construct an internal representation of the board and update its strategy based on past observations.

### L.11.13   POPGYM MINE SWEEPER

A partially observable version of the computer game Mine Sweeper, where the agent lacks direct visibility of the board. Observations include the coordinates of the most recently clicked tile and the number of adjacent mines. Clicking on a mined tile results in a negative reward and ends the game. To succeed, the agent must track previous selections and deduce mine locations based on the numerical clues, ensuring it avoids mines while uncovering safe tiles.

### L.11.14 POPGYM LABYRINTH EXPLORE

The environment consists of a procedurally generated 2D maze in which the agent earns rewards for reaching new, unexplored tiles. Observations are limited to adjacent tiles, requiring the agent to infer the larger maze layout through exploration. A small penalty per timestep incentivizes efficient navigation and discovery strategies.

### L.11.15 POPGYM LABYRINTH ESCAPE

This variation of Labyrinth Explore challenges the agent to find an exit rather than merely exploring the maze. The agent retains the same restricted observation space, seeing only nearby tiles. Rewards are only given upon successfully reaching the exit, making it a sparse reward environment where the agent must navigate strategically to achieve its goal.

## M   MIKASA-ROBO CUSTOMIZATION GUIDES

Code 7: ShellGameTouch: key difficulty knobs and a debug preset.

```python
# ShellGameTouch
from mani_skill.utils.registration import register_env
from mikasa_robo_suite.shell_game_touch import ShellGameTouchEnv
import gymnasium as gym

@register_env("ShellGameTouchDebug-v0", max_episode_steps=1000)
class ShellGameTouchDebugEnv(ShellGameTouchEnv):
    BALL_RADIUS = 0.02 # radius of the ball
    MIN_DIST = 0.2 # minimum distance between nearest cups
    TIME_OFFSET = 5 # how long the ball is visible (no cups)
    GOAL_THRESH = 0.08 # threshold for the goal

    env = gym.make("ShellGameTouchDebug-v0", num_envs=256, obs_mode="
        ↪ rgb", render_mode="all")
```

Code 8: ShellGamePush: key difficulty knobs and a debug preset.

```python
# ShellGamePush
from mani_skill.utils.registration import register_env
from mikasa_robo_suite.shell_game_push import ShellGamePushEnv
import gymnasium as gym

@register_env("ShellGamePushDebug-v0", max_episode_steps=1000)
class ShellGamePushDebugEnv(ShellGamePushEnv):
    BALL_RADIUS = 0.02 # radius of the ball
    MIN_DIST = 0.2 # minimum distance between nearest cups
    TIME_OFFSET = 5 # how long the ball is visible (no cups)
    GOAL_THRESH = 0.08 # threshold for the goal

    env = gym.make("ShellGamePushDebug-v0", num_envs=256, obs_mode="
        ↪ rgb", render_mode="all")
```

Code 9: ShellGamePick: key difficulty knobs and a debug preset.

```python
# ShellGamePick
from mani_skill.utils.registration import register_env
from mikasa_robo_suite.shell_game_pick import ShellGamePickEnv
import gymnasium as gym

@register_env("ShellGamePickDebug-v0", max_episode_steps=1000)
class ShellGamePickDebugEnv(ShellGamePickEnv):
    BALL_RADIUS = 0.02 # radius of the ball
    MIN_DIST = 0.2 # minimum distance between nearest cups
    TIME_OFFSET = 5 # how long the ball is visible (no cups)
    GOAL_THRESH = 0.08 # threshold for the goal

    env = gym.make("ShellGamePickDebug-v0", num_envs=256, obs_mode="
        ↪ rgb", render_mode="all")
```

## M.1 INTERCEPT

Code 10: Intercept: controlling projectile speed and target tolerance.

```python
from mani_skill.utils.registration import register_env
from mikasa_robo_suite.intercept import InterceptBaseEnv
import gymnasium as gym

@register_env("InterceptDebug-v0", max_episode_steps=1000)
class InterceptDebugEnv(InterceptBaseEnv):
    VELOCITY_RANGE = (0.0, 0.0)  # (min_v, max_v) - range for the ball
        ↪   velocity randomization
    BALL_RADIUS = 0.02  # radius of the ball
    GOAL_RADIUS = 0.1  # radius of the goal region

    env = gym.make("InterceptDebug-v0", num_envs=256, obs_mode="rgb",
        ↪ render_mode="all")
```

Code 11: InterceptGrab: grasp-based variant with velocity randomization.

```python
from mani_skill.utils.registration import register_env
from mikasa_robo_suite.intercept_grab import InterceptGrabBaseEnv
import gymnasium as gym

@register_env("InterceptGrabDebug-v0", max_episode_steps=1000)
class InterceptGrabDebugEnv(InterceptGrabBaseEnv):
    VELOCITY_RANGE = (0.0, 0.0)  # (min_v, max_v) - range for the ball
        ↪   velocity randomization
    BALL_RADIUS = 0.02  # radius of the ball

    env = gym.make("InterceptGrabDebug-v0", num_envs=256, obs_mode="
        ↪ rgb", render_mode="all")
```

## M.2 ROTATE

Code 12: RotateLenient: one- vs. two-sided rotation with tolerance control.

```python
from mani_skill.utils.registration import register_env
from mikasa_robo_suite.rotate_lenient import RotateLenientEnv
import gymnasium as gym

@register_env("RotateLenientDebug-v0", max_episode_steps=1000)
class RotateLenientDebugEnv(RotateLenientEnv):
    MODE = "pos_angle" # "pos_angle" or "pos_neg_angle" - defines
        ↪ possible directions of rotation
    PEG_HALF_WIDTH = 0.025 # peg half width
    PEG_HALF_LENGTH = 0.12 # peg half length
    ANGLE_THRESHOLD = 0.1 # (radians) defines the permissible
        ↪ deviation from the target angle

    env = gym.make("RotateLenientDebug-v0", num_envs=256, obs_mode="
        ↪ rgb", render_mode="all", angle_threshold=ANGLE_THRESHOLD)
```

Code 13: RotateStrict: stricter alignment for fine-grained control.

```python
from mani_skill.utils.registration import register_env
from mikasa_robo_suite.rotate_strict import RotateStrictEnv
import gymnasium as gym

@register_env("RotateStrictDebug-v0", max_episode_steps=1000)
class RotateStrictDebugEnv(RotateStrictEnv):
    MODE = "pos_angle" # "pos_angle" or "pos_neg_angle" - defines
        ↪ possible directions of rotation
    PEG_HALF_WIDTH = 0.025 # peg half width
    PEG_HALF_LENGTH = 0.12 # peg half length
    ANGLE_THRESHOLD = 0.1 # (radians) defines the permissible
        ↪ deviation from the target angle

    env = gym.make("RotateStrictDebug-v0", num_envs=256, obs_mode="rgb
        ↪ ", render_mode="all", angle_threshold=ANGLE_THRESHOLD)
```

## M.3 TAKEITBACK

Code 14: TakeItBack: goal-region and object-size controls.

```python
from mani_skill.utils.registration import register_env
from mikasa_robo_suite.take_it_back import TakeItBackEnv
import gymnasium as gym

@register_env("TakeItBackDebug-v0", max_episode_steps=1000)
class TakeItBackDebugEnv(TakeItBackEnv):
    GOAL_RADIUS:    float = 0.08  # radius of the goal region
    CUBE_HALFSIZE:  float = 0.02  # cube size

    env = gym.make("TakeItBackDebug-v0", num_envs=256, obs_mode="rgb",
        ↪  render_mode="all")
```

## M.4 REMEMBERCOLOR

Code 15: RememberColor: visibility window, occlusion delay, and tolerance.

```python
from mani_skill.utils.registration import register_env
from mikasa_robo_suite.remember_color import RememberColorBaseEnv
import gymnasium as gym

@register_env("RememberColor4Debug-v0", max_episode_steps=1000)
class RememberColorDebugEnv(RememberColorBaseEnv):
    COLORS        = 4      # 1-9 unique cubes
    TIME_OFFSET   = 200    # how long target cube is shown
    GOAL_THRESH   = 0.03   # more difficult goal threshold
    CUBE_HALFSIZE = 0.02   # cubes size
    DELTA_TIME    = 100    # empty table duration (seconds)

    env = gym.make("RememberColor4Debug-v0", num_envs=256, obs_mode="
        ↪ rgb", render_mode="all", delta_time=DELTA_TIME)}
```

## M.5 REMEMBERSHAPE

Code 16: RememberShape: number of shapes, scale, and fixed color option.

```python
from mani_skill.utils.registration import register_env
from mikasa_robo_suite.remember_shape import RememberShapeBaseEnv
import gymnasium as gym

@register_env("RememberShape6Debug-v0", max_episode_steps=1000)
class RememberShapeDebugEnv(RememberShapeBaseEnv):
    SHAPES        = 6      # 1-9 unique shapes
    TIME_OFFSET   = 200    # how long target cube is shown
    GOAL_THRESH   = 0.03   # more difficult goal threshold
    SHAPE_SCALE   = 0.02   # cubes size
    COLOR = [0, 0, 255, 255] # each object has the same color
    DELTA_TIME    = 100    # empty table duration (seconds)

    env = gym.make("RememberShape6Debug-v0", num_envs=256, obs_mode="
        ↪ rgb", render_mode="all", delta_time=DELTA_TIME)
```

## M.6 REMEMBERSHAPEANDCOLOR

Code 17: RememberShapeAndColor: composite cue space and timing.

```python
from mani_skill.utils.registration import register_env
from mikasa_robo_suite.remember_shape_and_color import
    ↪ RememberShapeAndColorBaseEnv
import gymnasium as gym

@register_env("RememberShapeAndColor4x1Debug-v0", max_episode_steps
    ↪ =1000)
class RememberShapeAndColorDebugEnv(RememberShapeAndColorBaseEnv):
    SHAPES        = 4 * 1 # 1-5 unique shapes with 3 colors (1-15
        ↪ combinations)
    TIME_OFFSET   = 200    # how long target cube is shown
    GOAL_THRESH   = 0.03   # more difficult goal threshold
    SHAPE_SCALE   = 0.02   # cubes size
    COLOR = [0, 0, 255, 255] # each object has the same color
    DELTA_TIME    = 100    # empty table duration (seconds)

    env = gym.make("RememberShapeAndColor4x1Debug-v0", num_envs=256,
        ↪ obs_mode="rgb", render_mode="all", delta_time=DELTA_TIME)
```

## M.7 BunchOfColors

Code 18: BunchOfColors: set size and presentation timing.

```python
from mani_skill.utils.registration import register_env
from mikasa_robo_suite.bunch_of_colors import BunchOfColorsEnv
import gymnasium as gym

@register_env("BunchOfColors6Debug-v0", max_episode_steps=1000)
class RememberShapeAndColorDebugEnv(BunchOfColorsEnv):
    COLORS        = 6      # 1-9 unique cubes
    GOAL_THRESH   = 0.03   # more difficult goal threshold
    CUBE_HALFSIZE = 0.02   # cubes size
    SEQUENCE_LENGTH = 15    # Number of cubes to show in sequence (1-9)
    STEP_DURATION =  15      # Duration to show each cube
    EMPTY_DURATION = 5      # Duration of empty table
    DELTA_TIME    = 100    # empty table duration (seconds)

    env = gym.make("BunchOfColors6Debug-v0", num_envs=256, obs_mode="
        ↪ rgb", render_mode="all", delta_time=DELTA_TIME)
```

## M.8 SeqOfColors

Code 19: SeqOfColors: sequence length and tempo.

```python
from mani_skill.utils.registration import register_env
from mikasa_robo_suite.seq_of_colors import SeqOfColorsEnv
import gymnasium as gym

@register_env("SeqOfColors6Debug-v0", max_episode_steps=1000)
class RememberShapeAndColorDebugEnv(SeqOfColorsEnv):
    COLORS        = 6      # 1-9 unique cubes
    GOAL_THRESH   = 0.03   # more difficult goal threshold
    CUBE_HALFSIZE = 0.02   # cubes size
    SEQUENCE_LENGTH = 15    # Number of cubes to show in sequence (1-9)
    STEP_DURATION =  15      # Duration to show each cube
    EMPTY_DURATION = 5      # Duration of empty table
    DELTA_TIME    = 100    # empty table duration (seconds)

    env = gym.make("SeqOfColors6Debug-v0", num_envs=256, obs_mode="rgb
        ↪ ", render_mode="all", delta_time=DELTA_TIME)
```

## M.9 ChainOfColors

Code 20: ChainOfColors: chained sub-episodes with controlled pace.

```python
from mani_skill.utils.registration import register_env
from mikasa_robo_suite.seq_of_colors import SeqOfColorsEnv
import gymnasium as gym

@register_env("SeqOfColors6Debug-v0", max_episode_steps=1000)
class RememberShapeAndColorDebugEnv(SeqOfColorsEnv):
    COLORS        = 6      # 1-9 unique cubes
    GOAL_THRESH   = 0.03   # more difficult goal threshold
    CUBE_HALFSIZE = 0.02   # cubes size
    SEQUENCE_LENGTH = 15    # Number of cubes to show in sequence (1-9)
    STEP_DURATION =  15      # Duration to show each cube
    EMPTY_DURATION = 5      # Duration of empty table
    DELTA_TIME    = 100    # empty table duration (seconds)

    env = gym.make("SeqOfColors6Debug-v0", num_envs=256, obs_mode="rgb
        ↪ ", render_mode="all", delta_time=DELTA_TIME)
```

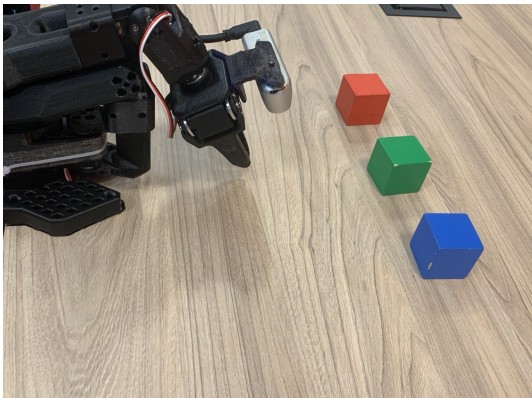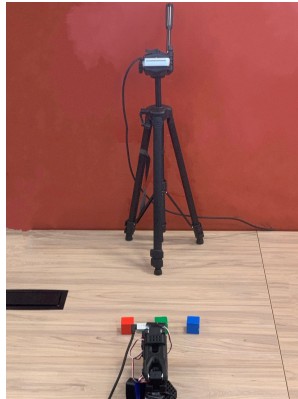

Figure 26: Real-world experimental setup for memory-intensive tabletop manipulation. The platform uses the SO-101 arm (Knight et al., 2024) equipped with a wrist-mounted RGB camera for egocentric observations and an external tripod-mounted RGB camera providing a fixed overhead view. The scene contains three colored cubic objects used in the `RememberColor3` family of tasks. This configuration matches the sensing and embodiment used for fine-tuning and evaluation in our real-world experiments.

Table 10: Real-world performance of the fine-tuned $\pi_{0.5}$ on the three evaluation tasks using 30 episodes per task. For each color and each task, we report the fraction of episodes where the robot touched the correct cube (`is_touched`) and where it successfully picked it (`is_picked`).

| Color | Task 1 | | Task 2 | | Task 3 | |
|---|---|---|---|---|---|---|
| | is_touched | is_picked | is_touched | is_picked | is_touched | is_picked |
| Total | 1.00 | 0.80 | 0.63 | 0.43 | 0.10 | 0.10 |
| Red | 1.00 | 0.90 | 0.70 | 0.40 | 0.10 | 0.10 |
| Green | 1.00 | 0.90 | 0.70 | 0.60 | 0.20 | 0.20 |
| Blue | 1.00 | 0.60 | 0.50 | 0.30 | 0.00 | 0.00 |

# N    REAL-WORLD EXPERIMENTS

We conducted a set of real-world experiments to evaluate whether modern VLA models can solve physical instances of the atomic memory-intensive tabletop manipulation tasks introduced in MIKASA-Robo benchmark. Although these tasks are visually simple, they impose explicit requirements on temporal information retention that current VLA architectures do not model. Our objective is to determine whether the failure modes observed in simulation persist under real-world sensing, actuation, and embodiment, and whether these failures can be attributed to memory limitations rather than to different confounding factors.

The experimental platform is shown in Figure 26. We used the SO-101 arm (Knight et al., 2024), the `lerobot` library (Cadene et al., 2024), and a pretrained $\pi_{0.5}$ model (Black et al., 2025) that we fine-tuned specifically for each of the three real-world tasks described below. The setup employs a wrist-mounted RGB camera for egocentric observations and a tripod-mounted RGB camera for a fixed overhead view. The objects used in all experiments are the three colored cubes from the `RememberColor3-v0` family of tasks.

Real-world experiments introduce additional sources of variation that can obscure the contribution of memory, such as contact inaccuracies, lighting changes, and pose drift. To isolate memory as the limiting factor, we designed a three-stage evaluation protocol. **Task 1** verifies that the demonstration pipeline, training process, and embodiment are sufficient for simple MDP problems. **Task 2** introduces dynamic scene changes without occlusion to test robustness to distribution shifts that do not require long-horizon memory. **Task 3** instantiates the full occluded long-horizon dependency of `RememberColor3-v0` to directly test memory.

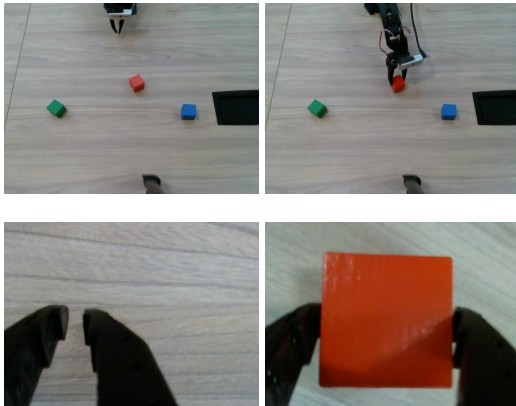

Figure 27: Task 1 (Sanity Check): real-world execution of the fully observable pick-and-place instruction. The robot observes all three cubes (red, green, blue) throughout the episode, receives the instruction to pick a specified color, and performs a standard MDP pick action without any memory requirement.

For each task, we collected a balanced dataset of 300 expert teleoperated demonstrations, with 100 trajectories for each target color. We then fine-tuned the pretrained $\pi_{0.5}$ model for 100,000 gradient steps using `lerobot`. Across all tasks we obtained 900 expert trajectories, which we will release in the camera-ready version. Quantitative results are summarized in Table 10.

**Task 1: Sanity Check.** This task verifies that the system can solve a fully observable pick-and-place instruction without any memory requirement. The robot observes three cubes on the table (red, green, blue) throughout the entire episode. Given the instruction "*Pick the {color} cube*", with {color} in {red, green, blue}, the robot must pick the corresponding cube. Since both the instruction and all objects remain visible at every timestep, the task is a standard MDP and demands no temporal memory. We collected 300 teleoperated demonstrations for fine-tuning (100 trajectories for each of the three target colors) and evaluated the fine-tuned $\pi_{0.5}$ across 30 episodes (10 per color). Since the correct cube is always visible, the task requires no temporal memory. Execution examples are shown in Figure 27. As reported in Table 10, the model reliably touches the correct cube in all episodes and achieves a pick success rate of 0.80. Per-color performance is consistent for red and green, with a lower pick success rate for blue, but the critical observation is that the touch success rate is 100 percent across all colors. This indicates that the VLA consistently identifies the correct target cube, and that the residual errors in Task 1 arise from grasp execution rather than from failures in interpreting the instruction or selecting the correct object. These results confirm that the training pipeline and the fine-tuning procedure are sufficient for standard, fully observed manipulation behaviors.

**Task 2: Dynamic Environment.** This task introduces motion and dynamic scene changes without creating a genuine memory requirement. The robot receives the instruction "*Remember the color of the cube and then pick the matching one*". It first observes a single cube of one color on the table. After this observation, we throw the remaining two cubes of other colors onto the table. The robot must pick the cube it observed initially. Although the instruction implies memory, the horizon between the initial view and the final decision is extremely short. With action chunking the model effectively retains the initial observation inside a single generated action sequence, so the task is solvable even without an explicit temporal memory mechanism. This task therefore controls for robustness to dynamic environmental changes while avoiding the occlusion and long horizon dependencies of `RememberColor3-v0`. Task 2 tests whether the model remains robust when the scene changes dynamically without creating a long-horizon dependency. A representative execution is shown in Figure 28. As summarized in Table 10, the model successfully touches the correct cube in 0.63 of episodes and completes the pick in 0.43. These results indicate that the model's performance degrades under dynamic scenes even though no actual memory is required. The model still succeeds substantially more often than in Task 3, which suggests that failures in Task 3 arise from missing memory rather than from difficulties with dynamic perception.

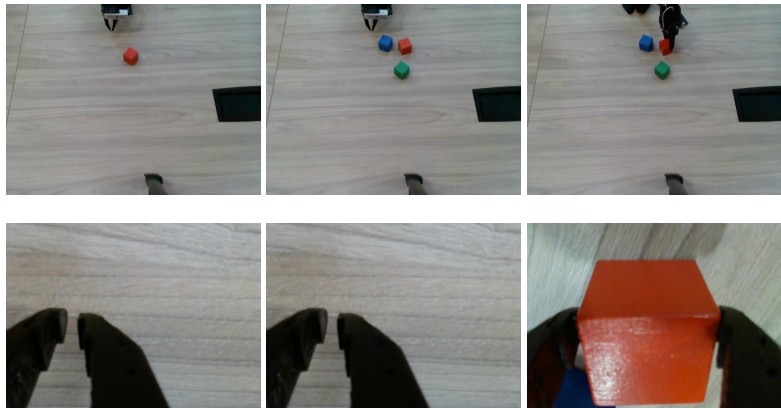

Figure 28: Task 2 (Dynamic Environment): real-world execution under short-horizon scene changes without occlusion. The robot first observes a single target cube, after which the remaining cubes are thrown into the scene, introducing dynamic motion and distribution shift while maintaining full visibility of all objects. The sequence illustrates the robot's approach and grasp using the wrist-mounted camera. Although the instruction suggests memory, the temporal gap is short enough that action chunking preserves the initial observation, which makes the task solvable without an explicit long-horizon memory mechanism.

**Task 3: Real World `RememberColor3-v0`.** Task 3 instantiates the full memory requirement. The robot first observes the target cube for several timesteps. We then occlude the target cube using a tablet, place the two remaining cubes underneath the tablet, permute the positions of all three cubes under occlusion, and finally reveal the scene. The robot must recall the color observed before occlusion and identify the matching cube in the post-shuffle configuration. Figure 29 illustrates a representative execution. As shown in Table 10, the robot's performance collapses in this setting, with only 0.10 touch success and 0.10 pick success overall, with similar patterns across individual colors. The drop in performance relative to Task 2 demonstrates that the model fails to retain the required color information across the occluded interval. Since conditions other than the long-horizon memory dependency are comparable between Task 2 and Task 3, the discrepancy isolates memory as the dominant failure mode.

**Discussion and Conclusion.** The comparison across Tasks 1, 2, and 3 in Table 10 demonstrates a systematic degradation as the memory requirement increases. The model performs reliably in a standard MDP setting (Task 1), begins to degrade under purely dynamic variation (Task 2), and fails almost entirely when long-horizon memory is required (Task 3). These findings align with our simulation results and provide strong evidence that current VLA models lack the temporal memory capabilities required for even simple real-world memory tasks. The results underscore the need for architectures with explicit and robust long-horizon memory mechanisms in real-world robot learning.

**Open-Source Datasets.** To ensure full reproducibility and to support further research on memory-intensive real-world manipulation, we publicly release all datasets used for fine-tuning and evaluation on Hugging Face. Each task contains 300 real-world trajectories in total, comprising 100 red, 100 green, and 100 blue cube episodes. The datasets are split differently across tasks solely for practical hosting and download convenience, while preserving the same color balance and total trajectory count.

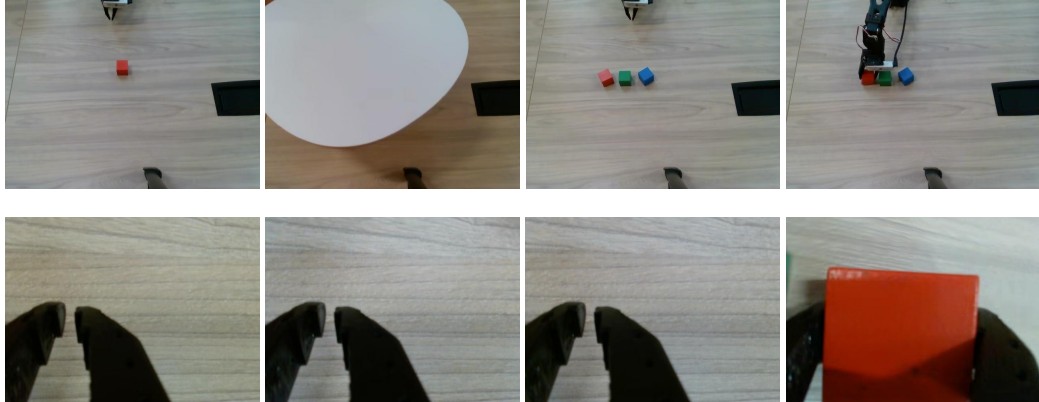

Figure 29: Task 3 (Real-world `RememberColor3-v0`): execution of the full long-horizon memory requirement with occlusion and object permutation. The robot first observes the target cube, after which the cube is covered with a tablet and all three cubes are shuffled under occlusion. Once the tablet is removed, the robot must identify and pick the originally observed cube despite the occluded interval and the unknown permutation. The sequence shows the initial observation, the occlusion event, the post-shuffle scene, and the eventual grasp attempt from the wrist-mounted camera view.

Table 11: Publicly released real-world datasets used in our experiments.

| Dataset | URL |
| --- | --- |
| *Task 1* | |
| so101_red_cube_100_task1 | https://huggingface.co/datasets/avanturist/so101_red_cube_100_task1 |
| so101_green_cube_100_task1 | https://huggingface.co/datasets/avanturist/so101_green_cube_100_task1 |
| so101_blue_cube_100_task1 | https://huggingface.co/datasets/avanturist/so101_blue_cube_100_task1 |
| *Task 2* | |
| so101_rgb_cube_300_task2 | https://huggingface.co/datasets/avanturist/so101_rgb_cube_300_task2 |
| *Task 3* | |
| so101_rgb_cube_300_task3_part_1_of_6 | https://huggingface.co/datasets/avanturist/so101_rgb_cube_300_task3_part_1_of_6 |
| so101_rgb_cube_300_task3_part_2_of_6 | https://huggingface.co/datasets/avanturist/so101_rgb_cube_300_task3_part_2_of_6 |
| so101_rgb_cube_300_task3_part_3_of_6 | https://huggingface.co/datasets/avanturist/so101_rgb_cube_300_task3_part_3_of_6 |
| so101_rgb_cube_300_task3_part_4_of_6 | https://huggingface.co/datasets/avanturist/so101_rgb_cube_300_task3_part_4_of_6 |
| so101_rgb_cube_300_task3_part_5_of_6 | https://huggingface.co/datasets/avanturist/so101_rgb_cube_300_task3_part_5_of_6 |
| so101_rgb_cube_300_task3_part_6_of_6 | https://huggingface.co/datasets/avanturist/so101_rgb_cube_300_task3_part_6_of_6 |

