# OpenReview forum: "Memory, Benchmark & Robots: A Benchmark for Solving Complex Tasks with Reinforcement Learning"
_ICLR.cc/2026/Conference — ICLR 2026 Poster_

### Official Review · Reviewer_KGih · 2025-10-29

**Soundness:** 3
**Presentation:** 3
**Contribution:** 2
**Rating:** 4
**Confidence:** 4

**Summary:**

The paper “Memory, Benchmark & Robots: A Benchmark for Solving Complex Tasks with Reinforcement Learning” introduces MIKASA (Memory-Intensive Skills Assessment Suite for Agents), a unified benchmark framework designed to systematically evaluate memory mechanisms in reinforcement learning (RL). Recognizing the absence of a standard evaluation suite for memory-based RL, especially in robotic manipulation, the authors propose a taxonomy of four memory types: object memory, spatial memory, sequential memory, and memory capacity. Building upon this taxonomy, they release two complementary components: MIKASA-Base, a consolidated collection of existing memory-intensive RL environments unified under a common API, and MIKASA-Robo, a new suite of 32 tabletop robotic manipulation tasks that require various forms of temporal and spatial memory. Experimental evaluations with online RL, offline RL, and VLAs demonstrate that current models, such as PPO-LSTM, Decision Transformer, and Octo, struggle to generalize across tasks with strong memory dependencies. The results highlight that while RL agents can perform well under full observability, their performance deteriorates sharply when memory retention is required, validating MIKASA as a rigorous benchmark for developing and assessing memory-centric RL algorithms.

**Strengths:**

The paper introduces a benchmark that systematically studies memory in reinforcement learning for tabletop manipulation, extending ManiSkill3 with new task variants emphasizing temporal dependency and spatial recall. The motivation is relevant and well-grounded, as memory remains a key bottleneck for long-horizon robotic reasoning. The paper provides clear organization, a well-documented codebase, and thorough evaluations across multiple RL and imitation learning baselines. By categorizing tasks based on distinct memory demands, the benchmark offers a structured framework for analyzing policy retention and forgetting behaviors. The authors also contribute useful diagnostic discussions and visualization tools for understanding agent performance over time. Overall, the work is timely and thoughtfully executed, offering a meaningful step toward standardized evaluation of memory mechanisms in robotic learning, even though its scope remains largely limited to simplified simulation settings.

**Weaknesses:**

There are some weaknesses arise:

1.This benchmark foucs on solving memory issuse encuntered in the table top manipulation scenarios. The benchmark is build upon maniskill3, where there is only a franka panda arm as the embodiment. And the manipulation task distribution is limited at pick-place blocks with various colors. However, in the real-world, more complex spatial resoning and operations are required. The lack of more diverse, real-world tasks and embodiments (such as retrieving a can of cola from a closed refrigerator, or a humanoid robot sequentially using both arms to assemble components collaboratively) undermines the overall contribution of this benchmark. Therefore, this benchmark can be considered merely an incremental supplement to the previous Maniskill3.

2.This benchmark does not conider the sim2real gaps that encounter in the real-world deployment, even though the authors claimed that their unified framework that can enable more robust systems for real-world use. Lack of real-world performance as evidence undermines the overall contribution of this benchmark. The RL memory problems that authors analyzed occur only in simulation or also in real-world? If memory problems (such as occuled objects, spatial reasoning) also encounter in the real-world, current evidence reported in the paper is not enough to support the claim about real-world application. Although I acknowledge simplfied simulation task setup is useful for controlled testing, these settings may not capture the sensory noise, continuous dynamics, and unstructured complexity found in real robot environments.

3.As this benchmark is an incremental work of maniskill3, current task distribution is limited in pick-place various blocks. However using such tasks for spatial/object evaluation is already covered in previous benchmarks, such as Ravens, LoHoRavens. Moreover, benchmark like LIBERO involve both object/spatial reasoning tasks and also include long-horizon tasks that require long-temporal dependencies, and the operated objects are more close to the real-world setting (e.g., coffee machine).

4.Although the benchmark divides tasks by memory type, the evaluation mainly reports success rates. It lacks diagnostic tools to analyze why an agent fails—whether due to perception errors, memory decay, or decision confusion. This makes it difficult to understand failure mechanisms for each memory type.

5.A minor point as the paper mentions that the evaluation of models like Octo and OpenVLA was restricted due to computational constraints, and fine-tuning was minimal. This means the reported performance may not reflect their full capability, reducing the fairness and depth of comparison across architectures. More extensive large-model adaptation is needed for a complete evaluation.

In summary, while the benchmark extends ManiSkill3 with 32 tasks, its contribution to memory-intensive RL remains limited because it does not address the sim-to-real gap and relies solely on success rates without richer diagnostic metrics. Most tasks center on pick-and-place cubes. Therefore, it is unclear how this benchmark provides substantial improvement over existing RL manipulation benchmarks.

**Questions:**

1.How well do the tabletop tasks in MIKASA-Robo reflect real-world robotic manipulation challenges? Since most tasks are simplified color or position memory exercises, can performance on these synthetic environments meaningfully predict how well an agent will perform in real, noisy physical settings?

2.The paper proposes four memory categories (object, spatial, sequential, capacity). How are the metrics and evaluation protocols designed to isolate these different memory mechanisms? Could overlapping factors (e.g., attention or perception) interfere with measuring pure memory ability?

3.MIKASA currently uses RGB and joint states for perception. Would incorporating other sensory modalities such as depth, tactile feedback, or language instructions change the evaluation of memory skills?

4.Has the author considered scenarios where occlusion exists from the outset (e.g., retrieving a bottle of cola from a closed refrigerator)? Can the task be accomplished solely with RGB+joint data in such cases? Additionally, can this benchmark support extensions like dual-arm coordination for humanoid robots and other heterogeneous embodied systems? Could the authors provide a unified interface to facilitate such expansions?

5.Have the authors considered adding diagnostic metrics or analyses beyond success rate, such as error decomposition or memory recall tracking, to better reveal the failure causes across different memory types?

6.The evaluation of large-scale vision-language-action models like Octo and OpenVLA was limited due to computational constraints. How might more thorough fine-tuning or longer training horizons change the conclusions about these models’ memory abilities? Does the benchmark currently support scalable training for such large architectures (provide scalable dataset or distributed training)?

I reserve the possibility of raising my score if the authors can adequately address all the concerns raised during the rebuttal phase.

---

> ### Author Response · Authors · 2025-11-22
> **Official comment by Authors**
>
> We thank Reviewer KGih for the careful assessment and appreciate the recognition of the benchmark’s structured focus on memory in tabletop manipulation and the clarity of the codebase and evaluations.
>
> > W1.
>
> We appreciate the Reviewer’s attention to embodiment and task diversity, but we strongly disagree with the assertion that MIKASA-Robo is restricted to “pick-place blocks with various colors” and that it constitutes only an incremental supplement to ManiSkill3. The Reviewer’s characterization does not reflect the actual task set, underlying design principles, or the conceptual contribution of the benchmark.
>
> First, the benchmark already includes a wide variety of object geometries and task families. The suite covers mugs, pegs, crosses, pyramids, tori, stars, balls, and other primitive objects across 32 tasks in 12 categories. These tasks capture object memory, spatial memory, sequential memory, and memory capacity. The allegation that the distribution is limited to colored blocks is factually inaccurate.
>
> Second, introducing additional object sets, including objects from YCB [1], BridgeData [2] or other commonly used datasets, is straightforward. The ShellGame task in our codebase explicitly demonstrates how to substitute custom meshes and properties. The benchmark design intentionally exposes this mechanism, and we provided an extended example in the Appendix B, Figures 8, 9 that shows how to adapt RememberColor3 to a custom-object version (RememberObject3). This directly responds to the Reviewer’s request for broader embodiments and object sets.
>
> Third, we intentionally use primitive objects. The goal of MIKASA-Robo is to isolate the memory dimension of robotic manipulation. Introducing complex meshes or delicate grasping conditions adds confounding factors that obscure the memory requirements we aim to diagnose. Primitive shapes yield clean, controllable environments where performance can be attributed to memory mechanisms rather than physics side effects or mesh-specific interactions.
>
> Fourth, tasks such as retrieving a can from a refrigerator or multistep humanoid assembly involve multiple simultaneous challenges. These include long-horizon contact sequences, object-specific grasping, and incidental failures unrelated to memory. Our design philosophy separates memory from orthogonal difficulties. If an agent cannot reliably solve the atomic memory tasks we introduce, there is no reason to believe it would succeed in complex composite scenarios. Conversely, if an agent succeeds on these isolated tasks, then large scale task suites such as BEHAVIOR-1K [3] provide natural downstream testbeds. What is absent in the field, and what MIKASA provides, is the set of atomic diagnostic tasks that allow controlled analysis before moving to full complexity.
>
> Fifth, memory requirements are independent of embodiment. The Franka Panda is one of the most widely adopted manipulation arms in RL and robotics research. It offers stable physics, high quality simulation assets, and broad community adoption. ManiSkill3, the underlying engine, supports alternative embodiments, and replacing Franka with another platform is simple. We have also added examples in the Appendix B, Figure 7 that show how to use other embodiments, addressing precisely the Reviewer’s concern.
>
> Finally, we strongly disagree with the characterization that our benchmark is an incremental extension of ManiSkill3. ManiSkill3 provides a simulation engine and a set of MDP tasks. Our contribution is a conceptual benchmark that provides a taxonomy of memory types, a systematic diagnostic design, and a comprehensive suite of manipulation tasks explicitly engineered to evaluate memory. The novelty lies in the conceptual structure, the four part taxonomy, the systematic coverage of memory types in robotic manipulation, and the unification of atomic memory tests in a domain where such diagnostic tasks did not exist. Also, the novelty lies in benchmarking a large number of policies from different paradigms: online RL, offline RL, IL, VLA. We use ManiSkill3 only as a backend physics engine. The benchmark’s contribution is orthogonal to the simulator and is not reducible to adding a few tasks to an existing platform. The Reviewer’s claim that this is an incremental supplement overlooks both the unified theory-driven structure and conceptual novelty of the provided tasks.
>
> We therefore maintain that MIKASA-Robo substantially advances the evaluation of memory in robotic manipulation and fills a gap that existing suites have not addressed.

---

> ### Author Response · Authors · 2025-11-22
> **Second part of the comment by Authors**
>
> > W2. sim2real, real world memory-intensive tasks
>
> Sim2real evaluation is outside the scope of this work. MIKASA is a diagnostic benchmark for memory, not an end to end sim2real system. The memory challenges we study, including occlusion, long horizon spatial recall, and sequential reasoning, occur directly in physical manipulation, and our tasks abstract these real phenomena into controlled settings.
>
> Developing such controlled simulation benchmarks without conducting full sim2real validation is standard practice in the community. Recent suites such as LIBERO [4] adopt similar simulation-only protocols, focusing on isolating specific algorithmic or cognitive capabilities rather than demonstrating real-world transfer. Our design aligns with this established methodology.
>
> Introducing sensory noise, clutter, or unstructured dynamics would confound the core variable under study. These factors are orthogonal to memory and would make it impossible to attribute failures to the agent’s memory mechanisms. The goal is to isolate memory, not to evaluate perception robustness or contact stability.
>
> If an agent cannot solve the atomic memory tasks in MIKASA, there is no justification to expect success in real world deployments where the same memory demands remain and additional complexities arise. MIKASA therefore provides an essential prerequisite for future sim2real studies, rather than attempting to replace them.
>
> > W3. Novelty
>
> While we appreciate the Reviewer’s perspective, the claim that our benchmark is an incremental extension of ManiSkill3 is incorrect. ManiSkill3 serves solely as a simulator backend. The contribution of our work lies in the conceptual memory taxonomy, the diagnostic design, and the 32 tasks explicitly engineered to isolate distinct memory mechanisms, none of which exist in ManiSkill3.
>
> We added Ravens, LoHoRavens, and LIBERO to Table 3. These benchmarks contain valuable manipulation tasks with spatial reasoning (not memory), but they do not provide controlled memory difficulty, nor do they disentangle memory from perception, grasping, or long horizon planning. Their tasks conflate multiple challenges, making it impossible to attribute failures to memory. This distinction is essential. These suites target spatial or object manipulation capabilities, not memory evaluation, so they are not directly comparable to MIKASA. They address a different problem class.
>
> MIKASA-Robo targets isolated memory phenomena. If an agent cannot solve these atomic tasks, there is no reason to expect success in more complex suites. Our benchmark therefore complements prior work by supplying the missing diagnostic foundation for evaluating memory in robotic manipulation.
>
> > W4. Evaluation metrics
>
> We thank the Reviewer for the suggestion, but the tasks are intentionally designed so that success rate is already a direct diagnostic of memory. All non memory factors are controlled: in state mode every task reaches 100 percent success, which confirms that perception and control are not limiting. When performance drops in RGB or RGB+joints mode, the failure is necessarily due to insufficient memory retention. The graded difficulty within each task type further pinpoints which memory capability breaks down. Additional analysis is possible through the provided oracle channels, but the benchmark’s purpose is to isolate memory so that success rates remain an interpretable and sufficient diagnostic metric. We also offer a variety of wrappers in the benchmark that can be used to track progression as you complete each task: rewards for approaching targets, scoring correctly, and so on (see Appendix A).
>
> > W5. VLA performance
>
> We thank the Reviewer for the comment and clarify that our evaluation of Octo and OpenVLA used LoRA rather than full model fine-tuning. We did not intend to imply that the evaluations were restricted in a way that limits the models’ capabilities. Prior work, including the original OpenVLA paper [5], shows that LoRA and full fine tuning achieve comparable performance while LoRA is substantially more efficient.
>
> > W6.
>
> We respectfully disagree. Our benchmark is not an extension of ManiSkill3. ManiSkill3 is only the simulator; the contribution is the memory taxonomy, the diagnostic design, and the 32 tasks explicitly constructed to isolate distinct memory mechanisms, which no prior manipulation benchmark provides. For more details, please, see our answers to W1, W2, W3, W4, W5.
>
> > Q1.
>
> The evaluation protocols isolate each memory category by removing confounds and ensuring that only the targeted memory skill is required. All tasks reach 100% success in state mode, confirming that control and perception are sufficient. In RGB modes, any degradation therefore reflects limited memory rather than attention or perception. Each task group scales difficulty along a single memory dimension, which clarifies the mechanism under test. Overlaps are minimized so performance aligns with the specific memory capability.

---

> ### Author Response · Authors · 2025-11-22
> **Third part of the comment by Authors**
>
> > Q3.
>
> Adding depth or tactile feedback would not change the evaluation of memory skills, since these modalities, like RGB, only encode the current timestep and cannot substitute for information that must be retained across the episode. Static language instructions used in current VLA models also do not aid memory, because they do not provide time varying cues about past events. In principle, dynamically generated language that explicitly summarizes past observations could assist memory-enhanced policies, but it would not replace the need for genuine memory mechanisms.
>
>
> > Q4.
>
> Tasks with initial occlusion, such as retrieving an item from a closed container, are feasible in principle, but they introduce additional difficulties unrelated to memory. Our benchmark intentionally avoids these confounders to isolate memory. RGB+joints mode is therefore sufficient for our atomic tasks, where the challenge is recalling previously observed information rather than inferring unseen content from the outset.
>
> The benchmark can be extended to dual arm or heterogeneous embodiments. ManiSkill3 already supports multiple embodiments, and MIKASA exposes all task logic independently of the robot model. A unified interface for such extensions is straightforward to provide, and we note that the revised Appendix B, Figure 7 now includes examples of adapting tasks to alternative embodiments.
>
> > Q5.
>
> We designed the tasks so that success rate is already an interpretable diagnostic of memory, because all non memory factors are controlled and each task is solvable at 100 percent in state mode. Failures in RGB based settings therefore reflect memory limitations rather than perception or control issues. More detailed analyses, such as error decomposition or recall tracking, depends on the policy architecture, not tasks. The benchmark supports this through oracle information channels, but the core objective is to isolate memory so that success rates remain a clean and sufficient diagnostic signal. Also, please, see our answer to W4 for other details.
>
> > Q6.
>
> As clarified in our response to W5, the evaluation of Octo and OpenVLA is not fundamentally limited. We used LoRA based adaptation, which prior work shows performs on par with full fine tuning for these architectures. MIKASA already provides 32 expert datasets, each with 1000 successful demonstrations, which supports fine tuning or RL based adaptation for large models. Scaling the training procedure beyond this is determined by the design of the VLA frameworks themselves, not by our benchmark.
> Longer training horizons would not change the conclusions. Current VLA models consume visual tokens rapidly and effectively operate on short temporal windows, so extending training does not address the absence of mechanisms for long term memory. The benchmark therefore already exposes the intrinsic memory limitations of these models.
>
> Please see the magenta-highlighted revisions in the updated .pdf of our paper (including Appendix).
>
> We thank Reviewer KGih for the thoughtful and constructive feedback. We believe we have addressed all raised questions and concerns, and we would be glad to engage further and respond to any additional points in continued discussion.
>
> [1] Calli, Berk, et al. "Benchmarking in manipulation research: The ycb object and model set and benchmarking protocols." arXiv preprint arXiv:1502.03143 (2015).
>
> [2] Walke, Homer Rich, et al. "Bridgedata v2: A dataset for robot learning at scale." Conference on Robot Learning. PMLR, 2023.
>
> [3] Li, Chengshu, et al. "Behavior-1k: A human-centered, embodied ai benchmark with 1,000 everyday activities and realistic simulation." arXiv preprint arXiv:2403.09227 (2024).
>
> [4] Liu, Bo, et al. "Libero: Benchmarking knowledge transfer for lifelong robot learning." Advances in Neural Information Processing Systems 36 (2023): 44776-44791.
>
> [5] Kim, Moo Jin, et al. "Openvla: An open-source vision-language-action model." arXiv preprint arXiv:2406.09246 (2024).

---

> ### Comment · Reviewer_KGih · 2025-11-24
>
> I appreciate the authors’ response. However, the lack of empirical analysis on real-world performance and occlusion-related scenarios leaves the reply insufficiently convincing. If the authors aim to position this work within robotics and embodied AI, real-robot performance and sim2real results are essential, as simulation-only evidence provides limited persuasive value. Therefore, I must retain my original score.

---

> > ### Author Response · Authors · 2025-12-02
> > **Real-world experiments**
> >
> > We thank the Reviewer for reiterating the request for real-robot experiments. We regret that, due to recent circumstances, we can only continue the discussion unilaterally. However, we would like to address the remaining points below. Our position remains that real-world testing is not required for the central scientific contribution of this work, which is a conceptual and diagnostic benchmark for evaluating an agent's memory. If a policy fails in controlled simulation settings where perception, physics, and embodiment are simplified, then there is no reason to expect it to succeed in the real world where the same memory requirements persist and additional sources of noise appear. Simulation is the necessary prerequisite for assessing memory, and failure in simulation already implies failure in real-world deployment.
> >
> > Nevertheless, we fully acknowledge the Reviewer’s interest in empirical validation on physical hardware. In response, we conducted a set of real-world experiments using the SO-101 robot and a fine-tuned $\pi_{0.5}$ model [1]. The setup is illustrated in Figure 26 (Appendix N). We designed three tasks that mirror the structure of MIKASA-Robo:
> >
> > **Task 1 (Sanity Check):** A fully observable pick and place instruction with no memory requirement. The model achieved 100 percent touch accuracy and 0.80 pick accuracy (Appendix, Table 10), which confirms that the fine-tuning pipeline and embodiment are sufficient for standard MDP tasks. See the Task 1 rollout in Appendix, Figure 27.
> >
> >
> > **Task 2 (Dynamic Environment):** The robot observes a single cube, then two additional cubes are dropped into the scene. This introduces dynamic variation but no long-horizon memory because the temporal gap is extremely short. The model reached 0.63 touch accuracy and 0.43 pick accuracy. The degradation relative to Task 1 indicates difficulties with dynamic perception but not with memory.  See the Task 1 rollout in Appendix, Figure 28.
> >
> >
> > **Task 3 (Real-World $\texttt{RememberColor3-v0}$):** The robot observes a target cube, the cube is occluded, two additional cubes are inserted under occlusion, and all three positions are permuted before the scene is revealed. This task reproduces the memory requirement of the MIKASA-Robo version. The model achieved only 0.10 touch accuracy and 0.10 pick accuracy, consistent across colors. The sharp drop from Task 2 to Task 3 isolates memory as the dominant failure mode, since perceptual and dynamical conditions are otherwise comparable.  See the Task 1 rollout in Appendix, Figure 29.
> >
> > Real-world performance of the fine-tuned $\pi_{0.5}$ on the three evaluation tasks using 30 episodes per task.  For each color and each task, the table reports the fraction of episodes where the robot touched the correct cube (`is_touched`) and where it successfully picked it (`is_picked`).
> >
> > | Color | **Task 1** is_touched | **Task 1** is_picked | **Task 2** is_touched | **Task 2** is_picked | **Task 3** is_touched | **Task 3** is_picked |
> > |----|-------|------|-----|----|--|-|
> > | **Total** | 1.00 | 0.80 | 0.63 | 0.43 | 0.10 | 0.10 |
> > | **Red**   | 1.00 | 0.90 | 0.70 | 0.40 | 0.10 | 0.10 |
> > | **Green** | 1.00 | 0.90 | 0.70 | 0.60 | 0.20 | 0.20 |
> > | **Blue**  | 1.00 | 0.60 | 0.50 | 0.30 | 0.00 | 0.00 |
> >
> > The real-world results therefore reproduce the same hierarchy observed in simulation: reliable performance under full observability, partial degradation under dynamic but non-occluded conditions, and near-complete failure when long-horizon occlusion is introduced. This alignment confirms that the failures diagnosed by MIKASA are genuine memory failures rather than artifacts of simulation fidelity, embodiment mismatch, or sensory noise.
> > For each of the three tasks, we collected a teleoperated dataset of 300 expert trajectories, balanced across colors with 100 trajectories per target color. This yields three datasets comprising a total of 900 trajectories, which we will release in the camera-ready version.
> >
> > We have added the real-robot experiments, figures, and quantitative results to the revised paper (Appendix N). We hope this fully addresses the Reviewer’s concerns.
> >
> > [1] Intelligence, Physical, et al. "$\pi_ {0.5} $: a Vision-Language-Action Model with Open-World Generalization." arXiv preprint arXiv:2504.16054 (2025).

---

### Official Review · Reviewer_YGtn · 2025-10-29

**Soundness:** 4
**Presentation:** 4
**Contribution:** 3
**Rating:** 8
**Confidence:** 3

**Summary:**

This paper proposes a comprehensive classification framework for memory-intensive RL tasks, integrating multiple previous tasks into a unified benchmark (MIKASA-Base). The authors also develop a novel benchmark of 32 memory-intensive tasks for tabletop robotic manipulation (MIKASA-Robo). They evaluate several current algorithms on their benchmark, demonstrating a clear need for the development of more effective algorithms for memory-intensive tasks.

Overall, I believe these contributions are of significant interest to the RL community and I strongly recommend acceptance. The comprehensive suite of tasks and thorough documentation make this a valuable resource for future research.

**Strengths:**

The paper is well-motivated, addressing an important gap in existing RL benchmarks. The comprehensive suite of 32 tasks provides comprehensive coverage of different types of memory requirements in physical robotic tasks.

The categorization of previous tasks and integration into a single framework is an important contribution that will facilitate further research.

The authors evaluate both online and offline algorithms and motivate the need for both in memory-intensive tasks.

The authors convincingly demonstrate that the tasks are challenging for current algorithms, showing the benchmark's value for driving future research.

The benchmark codebase is well-documented and the authors provide an easy startup notebook.

**Weaknesses:**

This is not the first benchmark for physical memory-intensive tasks. However, the authors acknowledge this limitation and clearly state that their benchmark captures more types of memory tasks than previous work.

The connection to cognitive processes and memory mechanisms could be more fully developed.

**Questions:**

**Questions for clarification:**

1. You mention that "The link between physical interaction and memory remains underexplored, motivating a framework for spatio-temporal memory in real-world tasks." What kinds of memory mechanisms do physical memory tasks require? What specific memory capabilities are not considered in the current framework? A more detailed discussion of this would strengthen the motivation.
2. What are your plans for continued support of the benchmark? Are you planning on adding new environments or algorithms? A section on future work would be welcome.
3. In Section 4.2, what does each memory task correspond to with regard to cognitive processes?

---

**Minor comments:**

Figure 6 is difficult to digest. I’m not sure what the legend items for Sequential Memory, Memory Capacity, Spatial Memory, and Object Memory correspond to on the plot since the background shading of the different tasks doesn’t match the colors in the legend. Additionally, Sequential Memory and Memory Capacity have different legend icons than Spatial Memory and Object Memory.

---

**Suggestions for improvement (not factored into score):**

1. The authors could elaborate more on the connection between cognitive processes and agentic memory requirements, maybe through a more systematic categorization. This would strengthen the theoretical foundation of the benchmark and help researchers understand what types of memory mechanisms are needed for different task categories.

---

> ### Author Response · Authors · 2025-11-22
> **Official comment by Authors**
>
> We thank Reviewer YGtn for the detailed assessment. We appreciate the recognition of the unified classification framework, the breadth of the 32-task robotic suite, and the value of evaluating both online and offline methods. The acknowledgement that the benchmark exposes clear limitations in current algorithms and that the codebase is accessible and well documented aligns closely with our goals for MIKASA as a resource for advancing research on memory-intensive RL.
>
> > W1. This is not the first benchmark for physical memory-intensive tasks. However, the authors acknowledge this limitation and clearly state that their benchmark captures more types of memory tasks than previous work.
>
> Thanks for raising this point. Prior work such as MemoryBench provides three spatial-memory tasks, but it covers only one memory type and does not support efficient large-scale parallelization.
>
> MIKASA is designed to fill this gap by offering a broader and more systematic suite: four categories of memory, 32 tasks in total, multiple (up to 3) difficulties for each task with easy customization (see Appendix B), varied observation spaces, and both dense and sparse rewards, all built on GPU-parallelized simulation. This makes it the first benchmark that enables comprehensive and scalable evaluation across the wide range of memory demands in robotics.
>
> > W2. The connection to cognitive processes and memory mechanisms could be more fully developed.
>
> Thanks for the comment. The paper already contains a dedicated section (Section 4.1. “Memory: From Cognitive Science to RL”) that connects our memory taxonomy to established cognitive science concepts and explains how these processes relate to memory demands in RL and robotics. This includes object permanence, categorical perception, working memory, memory span, causal reasoning, and transitive inference.
>
> If the Reviewer had a different connection in mind, we would welcome additional clarification so that we can adjust the manuscript as needed.
>
> > Q1. You mention that "The link between physical interaction and memory remains underexplored, motivating a framework for spatio-temporal memory in real-world tasks." What kinds of memory mechanisms do physical memory tasks require? What specific memory capabilities are not considered in the current framework? A more detailed discussion of this would strengthen the motivation.
>
> Thanks for raising this point. Physical manipulation tasks impose several concrete memory demands that differ from those in purely visual or grid-world settings. These include retaining object attributes across occlusions, maintaining spatial layouts under viewpoint changes, integrating long-horizon action-perception sequences, and handling distractors while preserving task-relevant information. Our four-category taxonomy captures these requirements through object memory, spatial memory, sequential memory, and memory capacity, which correspond to the dominant forms of temporal information retention needed in physical control.
>
> The benchmark does not attempt to exhaust all possible memory phenomena. Higher-level reasoning abilities, such as long-term episodic recall across tasks or meta-learning over task families, fall outside the present framework. These capabilities are orthogonal to the spatio-temporal demands we target and would require different problem formulations. However, investigating how agents handle spurious correlations and how memory is updated, rewritten, or forgotten under limited capacity is an important direction for future work and lies beyond the scope of the current framework. We have covered this in more detail in the Section 7 “Limitations and Future Work” in the updated version of the text.
>
> The Appendix, F “Memory Mechanisms in RL” already provides a brief overview of memory mechanisms used in RL, including recurrent architectures, state-space models, transformers, and graph-based approaches. The benchmark is designed so that these mechanisms can be evaluated systematically within the four memory categories above. In the updated version of the text, we have expanded this section by adding a discussion of other memory mechanisms: temporal convolutions, world models, external memory buffers, and autostigmergy.
>
> If the Reviewer had additional mechanisms or cognitive processes in mind, we would welcome clarification and would be glad to refine the manuscript further.
>
> > Q2. What are your plans for continued support of the benchmark? Are you planning on adding new environments or algorithms? A section on future work would be welcome.
>
> Yes, we will support the benchmark, since it is important for the community, and also we plan to add other tasks, including those related to spurious correlations and memory updates/rewrites (see our comment above), and we also plan to benchmark new VLA models, including the recently emerging memory-enhanced VLAs. We've covered this in more detail in Section 7.

---

> ### Author Response · Authors · 2025-11-22
> **Second part of the Author's comment**
>
> > Q3. In Section 4.2, what does each memory task correspond to with regard to cognitive processes?
>
> Thanks for the question. Section 4.2 already aligns each of the four memory task types with specific cognitive processes introduced in Section 4.1. To make this explicit:
>
> 1. Object memory corresponds to object permanence and categorical perception, since the agent must retain object identity and properties when they become unobservable.
>
> 2. Spatial memory corresponds to spatial working memory and mental map formation, requiring retention of layouts and previously seen locations.
>
> 3. Sequential memory corresponds to serial recall and working memory for ordered information, since the agent must remember and act on temporally structured sequences.
>
> 4. Memory capacity corresponds to memory span, since the agent must maintain multiple items simultaneously and handle interference from distractors.
>
> > Minor comments: Figure 6 is difficult to digest. I’m not sure what the legend items for Sequential Memory, Memory Capacity, Spatial Memory, and Object Memory correspond to on the plot since the background shading of the different tasks doesn’t match the colors in the legend. Additionally, Sequential Memory and Memory Capacity have different legend icons than Spatial Memory and Object Memory.
>
> Thank you for your comment, we have updated Figure 6 in the revised text.
>
> Please see the magenta-highlighted revisions in the updated .pdf of our paper.
>
> We are grateful to the Reviewer YGtn for their insightful comments. We hope we were able to address all questions and concerns, and we remain open to further discussion if you have any further comments.

---

> > ### Comment · Reviewer_YGtn · 2025-11-23
> >
> > I appreciate the authors' detailed response, including clarification on memory mechanisms, inclusion of future work, and updates to Figure 6. All of my concerns are addressed, and I continue leaning strongly toward acceptance.

---

> > > ### Author Response · Authors · 2025-11-26
> > > **Thank you**
> > >
> > > Thank you for the positive assessment and for supporting our work. We also appreciate your comments and suggestions, which helped improve the presentation and clarify the central aspects of the paper.

---

### Official Review · Reviewer_8hhi · 2025-10-30

**Soundness:** 3
**Presentation:** 2
**Contribution:** 3
**Rating:** 8
**Confidence:** 4

**Summary:**

This paper presents a benchmark for testing the capabilities of an agent's memory in the context of robotic manipulation with reinforcement learning (RL).
They distinguish different types of memory tasks as 1) object memory, 2) spatial memory, 3) sequential memory, and 4) memory capacity, which seems intuitive and, to the best of my knowledge, hasn't been done before.
They propose a set of Gymnasium-based tasks for memory-enhanced RL and a set of 32 robotic tasks.
They also release a set of offline visual-based datasets for sparse-reward manipulation tasks, for offline RL.
It's built upon ManiSkill3, which offers fast GPU parallelisation.
However, it is worth noting that while I do work on RL, I do not have experience with memory mechanisms in RL.

**Strengths:**

The proposed benchmark addresses spatial and temporal memory -- an important and timely topic in RL. The authors evaluate a broad set of baselines, and since none achieve strong performance across all tasks, the benchmark appears to have an appropriate level of difficulty, leaving room for future progress. It is built on the ManiSkill3 simulation framework, enabling fast GPU parallelisation -- a valuable design choice for scalability. Overall, the paper is well written and easy to follow, with only minor issues to fix.

**Weaknesses:**

- The main text lacks details on the quality of the offline RL datasets. The authors should explicitly state that these are expert datasets, as this information is important for readers and enables potential use of imitation learning methods.
- TD-MPC2 is cited as an arXiv preprint, but it has been published at ICLR 2024.
- The bibliography has inconsistent capitalisation (e.g., “Td-mpc2” should be “TD-MPC2”). This can be fixed by correctly using braces {} in BibTeX.
- It is unusual to use "Subsection". I would advise the authors to use "Section".
- Formatting issues:
    - Figure 1 needs more space around it.
    - Table 1 requires additional white space below it.
    - The text in Table 3 is too small; it should be made bigger.
    - The text in Figure 6 is too small and difficult to read; it should be enlarged.

**Questions:**

See weaknesses

---

> ### Author Response · Authors · 2025-11-22
> **Official comment by Authors**
>
> We thank Reviewer 8hhi for the careful evaluation. We value the recognition that the benchmark targets spatial and temporal memory in a principled way, offers a level of difficulty that exposes clear gaps in current methods, and benefits from the scalability provided by GPU-parallelized simulation.
>
> > W1. The main text lacks details on the quality of the offline RL datasets. The authors should explicitly state that these are expert datasets, as this information is important for readers and enables potential use of imitation learning methods.
>
> Thanks for the comment. We revised the text (see magenta highlight) to state clearly that the offline RL datasets consist of expert demonstrations and added an explicit note in the dataset description section to clarify their intended use for offline RL and imitation learning: “Training was performed using datasets consisting of 1000 successful trajectories per task, obtained by using PPO with oracle-level information about the environment (see details in Appendix C)”.
>
> > W2. TD-MPC2 is cited as an arXiv preprint, but it has been published at ICLR 2024.
>
> Thank you for your comment, we have corrected it in the text.
>
> > W3. The bibliography has inconsistent capitalisation (e.g., “Td-mpc2” should be “TD-MPC2”). This can be fixed by correctly using braces {} in BibTeX.
>
> Corrected in the text.
>
> > W4. It is unusual to use "Subsection". I would advise the authors to use "Section".
>
> Thanks, corrected in the text.
>
> > W5. Formatting issues: Figure 1 needs more space around it. Table 1 requires additional white space below it. The text in Table 3 is too small; it should be made bigger. The text in Figure 6 is too small and difficult to read; it should be enlarged.
>
> We appreciate your comments. We've adjusted the layout in the updated version of the paper.
>
> Please see the magenta-highlighted revisions in the updated .pdf of our paper.
>
> We would like to once again thank the Reviewer 8hhi for their comments regarding our work. We hope we were able to address all comments, and we look forward to further discussion if there are any remaining comments or questions.

---

> > ### Comment · Reviewer_8hhi · 2025-11-27
> >
> > Thanks for addressing my comments. The clarity of the paper has definitely improved.
> >
> > I have read the other reviews, rebuttals, and responses, and I continue to vote for acceptance.

---

> ### Author Response · Authors · 2025-11-29
> **Thank you**
>
> We thank you for the positive evaluation and for supporting the contribution of our work. Your comments and suggestions for improving the clarity of the paper were highly valuable.

---

### Official Review · Reviewer_iPbp · 2025-10-31

**Soundness:** 2
**Presentation:** 3
**Contribution:** 2
**Rating:** 6
**Confidence:** 3

**Summary:**

This paper introduces MIKASA (Memory-Intensive Skills Assessment Suite for Agents), a comprehensive benchmark suite aimed at systematically evaluating memory capabilities in reinforcement learning (RL) agents. The authors address a major gap in the field — the lack of standardized benchmarks to assess memory across diverse settings, particularly in tabletop robotic manipulation, where partial observability and temporal dependencies are critical.

**Strengths:**

1. The benchmark offers robust and comprehensive coverage, incorporating a diverse set of tasks that rigorously test models across a spectrum of memory requirements.

2. This benchmark successfully strikes a critical balance, bridging the gap between current, often over-simplified evaluation methods and the complexity of real-world, human-level tasks. By incorporating challenges that mirror the cognitive demands of human problem-solving while maintaining experimental tractability, it establishes a new, more relevant standard for assessing model performance

**Weaknesses:**

1. The benchmark is specialized for tabletop robotic manipulation tasks. While this area is critical for memory in robotics, the benchmark does not cover other key areas of robotic memory, such as continual learning where the agent must adapt to changing task distributions or physical environments over extended, sequential periods.

2. By design, the tasks are complex and memory-intensive, leading to high computational demands for training. This can limit the accessibility of the benchmark to researchers without substantial computational resources.

**Questions:**

1. How do you envision extending the benchmark to evaluate memory mechanisms in continual or lifelong learning settings, where robots must adapt to evolving task distributions or environments while not forgetting previous ones?

2. Given that the benchmark’s tasks are both complex and memory-intensive, have you explored ways to reduce the computational cost? How do you balance the benchmark’s realism and difficulty with inclusivity for a wider research community?

---

> ### Author Response · Authors · 2025-11-22
> **Official comment by Authors**
>
> We thank Reviewer iPbp for the detailed comments. We especially appreciate the recognition of the breadth and rigor of the benchmark and the acknowledgment that the suite offers a meaningful progression toward evaluating agents under conditions that more closely reflect human-level memory demands.
>
> > W1, Q1. The benchmark is specialized for tabletop robotic manipulation tasks. While this area is critical for memory in robotics, the benchmark does not cover other key areas of robotic memory, such as continual learning where the agent must adapt to changing task distributions or physical environments over extended, sequential periods.
> How do you envision extending the benchmark to evaluate memory mechanisms in continual or lifelong learning settings, where robots must adapt to evolving task distributions or environments while not forgetting previous ones?
>
> Continual and lifelong learning settings are important directions, but they are not part of the scope we target in this work. The benchmark focuses deliberately on episodic, controlled evaluations of memory rather than nonstationary or task-stream settings. That said, the environments in MIKASA are fully parameterized and documented in detail in the Appendix, and these specifications make it straightforward for others to construct continual or lifelong variants if desired by combining tasks and extending horizons. Our goal in this paper is to establish a rigorous foundation for memory evaluation in episodic manipulation, and the extensibility of the API allows researchers to build continual learning protocols on top of this foundation without modifying the core benchmark.
>
> > W2, Q2. By design, the tasks are complex and memory-intensive, leading to high computational demands for training. This can limit the accessibility of the benchmark to researchers without substantial computational resources.
> Given that the benchmark’s tasks are both complex and memory-intensive, have you explored ways to reduce the computational cost? How do you balance the benchmark’s realism and difficulty with inclusivity for a wider research community?
>
> Thank you for pointing this out. Our use of “memory-intensive” follows the definition in Section 3.2 “Memory-Intensive Environments” and refers to the algorithmic and behavioral demands on the agent’s internal state, not to higher hardware or runtime requirements.
>
> Concretely, Section 3.2 defines memory-intensive environments as those in which successful policies must retain and manipulate information over extended temporal windows under partial observability (for example, tracking occluded objects, recalling earlier instructions, or integrating multiple frames), rather than those that require larger models or more GPU RAM. Under this definition, MIKASA-Robo does not add computational overhead compared to standard visuomotor robotic benchmarks. For instance, a base PPO policy in our tasks has essentially the same compute and memory footprint as in ManiSkill3 [1], since observation and action spaces, simulator complexity, and rollout lengths are comparable and we do not mandate any additional architectural modules.
>
> In terms of accessibility, our design keeps episode lengths moderate and uses object sets and scene configurations that are on par with existing tabletop manipulation suites. The difficulty comes from temporal credit assignment and memory demands, not from heavier simulation or larger networks. In addition, the benchmark is intentionally scalable: following guides in Appendix B, users can reduce horizon length, number of distractors, or task diversity to obtain lighter variants without changing the underlying memory type.
>
> In summary, Section 3.2’s notion of “memory-intensive” is about “cognitive load” on the agent, not computational load on the hardware, which allows us to maintain realism in the memory requirements while keeping the benchmark accessible to groups with standard compute budgets.
>
> We thank Reviewer iPbp for the time and attention devoted to this feedback. We believe the responses above address all raised points
>
>
> [1] Tao, Stone, et al. "Maniskill3: Gpu parallelized robotics simulation and rendering for generalizable embodied ai." arXiv preprint arXiv:2410.00425 (2024).

---

### Author Response · Authors · 2025-11-26
**Gerenal Response by Authors**

We sincerely appreciate the Reviewers’ thoughtful evaluations and their recognition of our benchmark as a meaningful and timely contribution to the study of memory in robotics. Their feedback affirms the motivation, design choices, and empirical rigor of the work. Below, we summarize the key strengths highlighted across the reviews:
1. **Comprehensive and well-motivated benchmark design**
- Broad and diverse task suite that rigorously probes a wide spectrum of memory requirements, addressing a clear gap in existing RL benchmarks (iPbp, YGtn).
- Tasks reflect human-level cognitive demands while remaining experimentally tractable, establishing a more realistic standard for memory evaluation (iPbp).
- Clear motivation grounded in the importance of temporal and spatial memory for long-horizon robotic control (KGih).


2. **Structured taxonomy of memory types**
- Introduction of an intuitive categorization covering object, spatial, sequential, and capacity-based memory demands, offering a principled framework not systematically presented before (8hhi).
- Integration and unification of previously scattered task types into a single coherent framework that supports future research (YGtn, KGih).


3. **Robust empirical evaluation and difficulty calibration**
- Thorough assessment across online RL, offline RL, and IL baselines, demonstrating consistent task difficulty and room for future methodological progress (8hhi, YGtn, KGih).


- Convincing evidence that current algorithms struggle in different ways across tasks, reinforcing the benchmark’s value for developing improved memory mechanisms (YGtn).


4. **Implementation quality and usability**
- Well-documented codebase built on ManiSkill3, enabling fast GPU parallelization and scalable experimentation (8hhi, KGih).
- Clear paper organization and an easy-to-use starter notebook that lowers the barrier to adoption (YGtn, KGih).


We thank all Reviewers for their detailed assessments and constructive feedback. We have carefully revised the manuscript to address all raised questions, concerns, and presentation issues. Below we provide a structured summary of the key modifications and clarifications.

**Revisions to the Text**
1. Increased whitespace and adjusted spacing around figures and tables to improve overall readability.
2. Enlarged **Figure 6** and improved its layout and color scheme for clearer visual interpretation.
3. Added a detailed explanation of how the expert datasets were generated (**Section 6.3**, highlighted in magenta).
4. Included an expanded **Limitations and Future Work** section discussing spurious correlations, dynamic memory overwriting/forgetting, and broader implications (**Section 7**, magenta highlighted).
5. Added new demonstrations and analysis on **embodiment flexibility** (**Appendix B, Figure 7**).
6. Added new demonstrations illustrating how to extend atomic tasks to **realistic object manipulation** (**Appendix B, Figure 8**).
7. Added discussion and demonstrations on **cluttered scene construction** and its impact on task complexity (**Appendix B, Figure 9**).
8. Substantially expanded the **"Memory Mechanisms in RL"** section to include temporal convolutions, world models, external memory buffers, and autostigmergy (**Appendix F**, magenta highlighted).

---

We sincerely thank the Reviewers once again for their insightful comments. We believe that the revised paper fully addresses the concerns raised and significantly improves the clarity, completeness, and contribution of the paper.

---

### Author Response · Authors · 2025-12-02
**Real-World Experiments**

Reviewer KGih raised the request for real-robot evaluation. While the **core contribution of MIKASA is a diagnostic benchmark and does not require sim-to-real validation**, we have **added a real-world study** to the revised paper for completeness.

The new **Appendix N** contains:
1. **Experimental setup** with the SO-101 robot (Figure 26).
2. **Three real-robot tasks** mirroring MIKASA-Robo:

    2.1. **Task 1 (Fully observable pick-and-place)**.

    2.2. **Task 2 (Dynamic environment without occlusion)**.

    2.3. **Task 3 (Real-world $\texttt{RememberColor3-v0}$)** with occlusion and permutation.

3. **Quantitative results** (Table 10) showing the same performance hierarchy as in simulation.
4. **Rollout visualizations** for each task (Figures 27–29).
5. **A new teleoperated dataset** of `900 expert trajectories` that will be released with the camera-ready version.

These results confirm that the failure patterns observed in simulation persist on real hardware and arise from memory limitations rather than perception or embodiment artifacts.

---

### Meta-Review · Area_Chair_6h87 · 2026-01-09

**Summary:**

Reviewers generally valued the paper’s contribution as a well-designed benchmark for evaluating memory in RL and robotic manipulation, highlighting the clear memory taxonomy and the breadth of tasks (8hhi, YGtn). The main concerns shaping the decision were the benchmark’s scope and positioning relative to existing suites, the reliance on simulation for a robotics-facing contribution, and whether the evaluation provides sufficient validation beyond controlled settings (KGih, iPbp).

**Reviewer Concerns:**

Most presentation and clarity issues were resolved through revisions clarifying expert datasets, correcting citations, and improving figures and tables (8hhi). Conceptual questions about cognitive grounding, future support, and figure interpretability were addressed and judged resolved (YGtn), and scope concerns about continual learning and compute were mitigated by clear scoping and extensibility arguments (iPbp). The main remaining disagreement concerns robotics validation: despite added real-robot experiments showing consistent memory-related failures under occlusion, one reviewer maintained that broader real-world validation is still required (KGih).

**Reviewer Scores:**

Reviewer (8hhi) would likely maintain an accept score around 8, as their requested clarifications and formatting fixes were implemented and they explicitly reaffirmed acceptance.

Reviewer (YGtn) would likely maintain an accept score around 8, since their conceptual and presentation concerns were addressed and they stated continued strong acceptance.

Reviewer (iPbp) would plausibly increase slightly from 6 to around 6–7, given the clarified stance on compute/accessibility and the clear scoping of continual learning as future extensible work rather than a flaw.

Reviewer (KGih) would likely remain at 4 (or at most move to a borderline 5) because their central criterion emphasized real-robot/sim2real and broader task realism.

---

### Decision · Program_Chairs · 2026-01-26

Accept (Poster)